# Evaluating Neuron Explanations: A Unified Framework with Sanity Checks

**Tuomas Oikarinen** [1]  **Ge Yan** [1]  **Tsui-Wei Weng** [2]

## Abstract

Understanding the function of individual units in a neural network is an important building block for mechanistic interpretability. This is often done by generating a simple text explanation of the behavior of individual neurons or units. For these explanations to be useful, we must understand how reliable and truthful they are. In this work we unify many existing explanation evaluation methods under one mathematical framework. This allows us to compare existing evaluation metrics, understand the evaluation pipeline with increased clarity and apply existing statistical methods on the evaluation. In addition, we propose two simple sanity checks on the evaluation metrics and show that many commonly used metrics fail these tests and do not change their score after massive changes to the concept labels. Based on our experimental and theoretical results, we propose guidelines that future evaluations should follow and identify a set of reliable evaluation metrics.

## 1. Introduction

Deep neural networks (DNNs) have achieved great success on a wide range of tasks, but they are very difficult to understand and often perceived as black-boxes (Rudin et al., 2022). To address this challenge, the field of mechanistic interpretability has recently emerged, aiming to provide a clearer understanding of the internal mechanisms of DNNs.

Providing natural language explanations for small components of a neural network is an important part of mechanistic interpretability. Classic work in this area includes Network Dissection (Bau et al., 2017) and other works explaining individual neurons in deep vision models (Mu & Andreas, 2020; Hernandez et al., 2022; Oikarinen & Weng, 2023; Bai et al., 2025). Other examples include automated neuron explanations (Bills et al., 2023; Lee et al., 2023) for large language models, as well as explaining features of sparse autoencoders (Bricken et al., 2023; Templeton et al., 2024).

Despite the introduction of various approaches for generating neuron explanations, existing papers often use very different metrics to evaluate how good their descriptions are, and it is not clear how they compare to each other. In addition, many evaluation metrics have problems, as shown by (Huang et al., 2023). To ensure that unit explanations are reliable and trustworthy, it is crucial to establish a standardized framework for evaluation.

Motivated by the need for a standardized approach, in this work we unify many existing evaluation methods under a single mathematical framework: *NeuronEval*, which provides much needed conceptual clarity to the topic of explanation evaluation. This framework allows us to clearly compare and contrast current evaluation techniques and provides a more transparent understanding of the evaluation pipeline. Inspired by this, we introduce two sanity tests to validate the metrics, revealing that most commonly used evaluation metrics fail at least one of these basic tests and should not be relied on alone. In summary, in this paper we:

- Formalize the task of evaluating individual unit explanations and unify 20 existing methods under the *NeuronEval* framework, which lets us apply standard statistical methods to evaluating neuron explanations.

- Propose two sanity checks for evaluation metrics: *Missing Labels Test* and *Extra Labels Test*. We then test different metrics both theoretically and empirically across hidden layer and final layer neurons, as well as on linear probes on both vision and language models.

- Out of the 18 metrics tested only the following pass the tests: *Correlation*(Pearson), *Cosine Similarity*, *AUPRC*, *IoU* and *F1-score*. In contrast, many commonly used evaluations fail these sanity tests, such as: using biased *top-and-random* sampling, only evaluating on highly activating inputs(*Recall*), measuring *AUC* or mean activation difference *MAD*, and cannot be relied on by themselves.

[1]CSE, UC San Diego, CA, USA [2]HDSI, UC San Diego, CA, USA. Correspondence to: Tuomas Oikarinen <toikarinen@ucsd.edu>, Tsui-Wei Weng <lweng@ucsd.edu>.

*Proceedings of the 42$^{nd}$ International Conference on Machine Learning*, Vancouver, Canada. PMLR 267, 2025. Copyright 2025 by the author(s).

Our code and results are publicly available at https://github.com/Trustworthy-ML-Lab/Neuron_Eval.

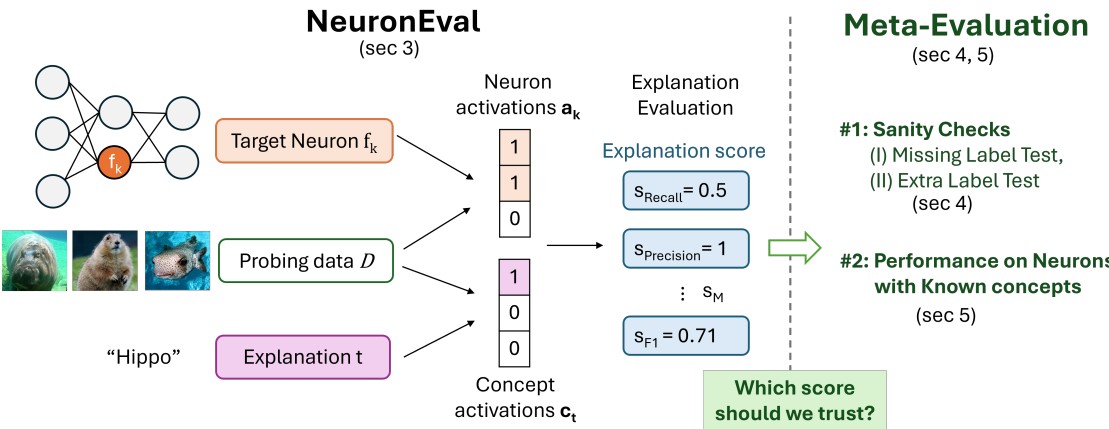

*Figure 1.* An overview of contributions. We first unify many existing explanation evaluation methods into a single mathematical framework *NeuronEval* containing 18 different metrics $s_M$. Next, we perform meta-evaluation via *NeuronEval* to answer the question: which evaluation metrics reliably measure how good an explanation is?

## 2. Definitions

### 2.1. What is an individual unit in a neural network?

In this paper we are focused on **individual unit** explanations. By a unit, we mean *a smaller part of a neural network that can have an independent meaning*. The simplest form of such units is a single neuron, or a single channel of a Convolutional Neural Network (CNN), but a unit can be any scalar function of network inputs. Other interesting units that fit in our framework are linear combinations of neurons (i.e. directions in activation space), which are considered to correspond to a specific interpretable concept. These are used in studies such as TCAV (Kim et al., 2018), Concept Bottleneck Models (Koh et al., 2020; Yuksekgonul et al., 2023; Oikarinen et al., 2023; Srivastava et al., 2024), Linear Probing (Gurnee et al., 2023) or steering vectors (Subramani et al., 2022). Finally, a unit could be a feature of a Sparse Autoencoder (Cunningham et al., 2023; Bricken et al., 2023) trained to disentangle a layer's activations into interpretable individual components. In this paper, we focus on units with scalar activations, excluding larger components e.g. attention heads.

### 2.2. Problem Description

Our focus in this paper is explanation[1] *evaluation*. In evaluation, a text explanation $t$ for a neuron $k$ is given, and our goal is to evaluate how faithfully this explanation describes the neuron. Below we define evaluation as a function $\mathcal{E}$:

$$\textbf{Evaluation (of Explanation } t) : \mathcal{E}(f_k^{0:l}, \mathcal{D}, t) \to \mathbb{R}$$

Here $\mathcal{E}$ is a function that takes a probing dataset $\mathcal{D}$, a

neural network[2] $f$ and a text description $t$, and returns a scalar score, where a higher score indicates the description is better i.e. more reliable/faithful. See Figure 1 for an illustration of the evaluation process.

For explanations to be reliable, evaluating explanations is very important, but the community has not reached an agreement on which $\mathcal{E}$ to use. Currently different papers often use different $\mathcal{E}$ without theoretical justifications. Thus, one focus of our paper is to perform *meta-evaluation* and rigorously investigate different evaluation metrics $\mathcal{E}$.

### 2.3. What are the goals of an explanation?

Neuron explanations are typically generated to improve our understanding of the model, which can then help improve, for example, safety and reliability of the models. However, this goal is vague and hard to measure generally. We believe a more precise definition of the goals of an explanation is essential for thinking clearly about how to evaluate them.

In this paper, we focus on **Input-based** neuron explanations, which make up the majority of existing explanations. We argue that the goal of an explanation is to provide a human-interpretable approximation of the following function:

$$\textbf{Input} \to \textbf{Unit Activation: } x \to f_k^{0:l}(x)\textbf{.}$$

A good explanation should be able to describe which inputs cause a high unit activation, and which do not. Note that there are other type of explanations being proposed (e.g. output-based, see App. B.2 for discussion.), which is not the focus of this work.

---

[1]Note: Throughout the paper we use "explanation", "concept", and "description" interchangeably to refer to a text description.

[2]We can write neuron $k$ in layer $l$ of neural network $f$ as a function $f_k^{0:l}(x)$, where $f^{i:j}$ represents the $i$ through $j$'th layers of the neural network $f(x)$. Here we assume the units are neurons for notational simplicity.

# 3. A Unified Evaluation Framework

We start by studying some popular neuron explanation evaluations in Sec. 3.1 and showing they can be formalized under the same notation. We then take a further step in Sec 3.2 to show that almost all existing work can be unified under this evaluation framework, which we call *NeuronEval*.

## 3.1. Formalizing Example Evaluations

**Example 1: Crowdsourced Highly Activating Inputs Evaluation.** Currently, perhaps the most common way to evaluate neuron explanations is to show (crowdsourced) human raters examples of highly activating inputs, and ask the raters whether these inputs can be described by the explanation (also known as 'concept') (Zhou et al., 2015; Bau et al., 2017; Oikarinen & Weng, 2023).

To analyze this approach more rigorously, we describe it through the following mathematical formulation: Given a set of probing data $\mathcal{D} = \{x_i\}$, we first define the **neuron activation vector** $a_k$ for neuron $k$ as:

$$a_k \in \mathbb{R}^{|\mathcal{D}|}, \ [a_k]_i = f_k^{0:l}(x_i) \ \forall i \in [1, |\mathcal{D}|]. \quad (1)$$

Here, the $i$-th entry of $a_k$ denotes the neuron activation on $i$-th input $x_i$. To represent highly activating inputs, we define a binarization function $B : \mathbb{R}^n \to \{0,1\}^n$, that takes a real vector and turns its element into 1 if it is a "high" activation and 0 otherwise. In particular, we use the $\text{top}_\alpha$ binarization function for neuron activations, where the activations in top $\alpha$ percentile return 1, and the rest return 0 as this is how highly activating inputs are typically selected. See Appendix A.1 for more detailed description.

Next, we describe the human ratings for a presence of text concept $t$ in the inputs by defining the **concept activation vector** $c_t$:

$$c_t \in \mathbb{R}^{|\mathcal{D}|}, \ [c_t]_i = \mathbb{P}(t|x_i) \ \forall i \in [1, |\mathcal{D}|] \quad (2)$$

Here each element of $c_t$, i.e. $[c_t]_i$, represents the fraction of raters that agreed on concept $t$ being present on that input $x_i$. Finally, given $a_k$ and $c_t$, we denote $s$ as an explanation score to quantify whether neuron $k$ is well explained by the concept $t$. In this evaluation, the fraction of highly activating inputs that have concept $t$ present is used to represent the quality of explanations. To formalize this we can define $s$ in terms of $a_k$ and $c_t$ as follows:

$$s := \mathbb{P}(\text{concept}|\text{neuron is active}) = \frac{B(a_k) \cdot B(c_t)}{||B(a_k)||_1} \quad (3)$$

since the number of highly activated inputs can be computed as $||B(a_k)||_1$ and the number of highly activated inputs that have concept $t$ present is $B(a_k) \cdot B(c_t)$. Note the value of $c_t$ on not highly activating inputs doesn't affect the score so we

don't need to evaluate it on all inputs. Here we also binarize the concept vector $c_t$ by rounding to 1 for elements $\geq 0.5$ (indicating concept probability $\geq 0.5$) and to 0 otherwise.

If we consider the goal of an explanation to be predicting neuron activations given whether the concept is present, i.e. *simulation*, then we can formulate the neuron explanation as a binary classification problem. Specifically, as we show in Appendix A.2 that Equation (3) is actually equivalent to measuring "Recall":

$$s_{\text{Recall}} = \frac{TP}{TP + FN} = \frac{B(a_k) \cdot B(c_t)}{||B(a_k)||_1} \quad (4)$$

So crowdsourced evaluation of highly activating inputs is equivalent to only measuring "Recall", i.e. we have $s = s_M$ with metric $M = \text{Recall}$.

**Example 2: IoU with Labeled Data.** In addition to crowdsourcing, other types of evaluations have been used for validating neuron explanations. For example, (Bau et al., 2017) collected a curated labeled dataset to evaluate neuron explanations by measuring the *Intersection over Union* (IoU, also known as Jaccard index) between neuron activations and concept labels on the annotated dataset. This measures $\mathbb{P}(\text{concept AND neuron is active})$ / $\mathbb{P}(\text{concept OR neuron is active})$.

We can also describe this evaluation with the same formalism as in **Example 1**, but *with two main changes*. First, we use a different *concept source*, where the elements of $c_t$ come from the labeled data instead of crowdsourced ratings. Second, we set the metric $M$ as IoU, giving us the following evaluation score:

$$s_{\text{IoU}} = \frac{B(a_k) \cdot B(c_t)}{||B(a_k)||_1 + ||B(c_t)||_1 - B(a_k) \cdot B(c_t)} \quad (5)$$

**Example 3: Automated Simulation Evaluation with Language Models.** More recently, (Bills et al., 2023) evaluate neuron explanations of large language models by measuring the correlation between actual neuron activations and activations predicted by a language model given the explanation.

While in different domains (Examples 1 and 2 are from vision models), this evaluation can also be formalized in the same manner as Example 1 and 2. First, the *concept source* for $c_t$ is now model predicted neuron activations scaled 0-1, as these are essentially pseudo-labels for the presence of the concept in the given input. Second, for the metric $M$, we now set it to be the Pearson's Correlation coefficient and obtain the evaluation score $s_{\text{Correlation}}$ as:

$$s_{\text{Correlation}} = \frac{1}{|\mathcal{D}|} \frac{\sum_i([a_k]_i - \mu(a_k)) \cdot ([c_t]_i - \mu(c_t))}{\sigma(a_k)\sigma(c_t)} \quad (6)$$

where $\mu(z)$ and $\sigma(z)$ is the mean and standard deviation of the vector $z$.

| Metric $M$ | Study | Concept Source $c_t$ | Granularity | Domain |
|---|---|---|---|---|
| | *NeuronEval*: A General Framework to Evaluate Neuron Explanations | | | |
| Recall | (Zhou et al., 2015) | Crowdsourced | Whole Input | Vision |
| $\sim$Recall | (Bau et al., 2017; Oikarinen & Weng, 2023) (Oikarinen et al., 2023; Bai et al., 2025) | Crowdsourced | Whole Input | Vision |
| Precision | (Srinivas et al., 2025) | Generative | Whole Input | Vision |
| F1-score | (Huang et al., 2023) (Gurnee et al., 2023) | Generative + Model Labeled data | Whole Input Per-token | Language Language |
| IoU | (Bau et al., 2017; Mu & Andreas, 2020) (La Rosa et al., 2024) | Labeled data | Per-pixel | Vision |
| Accuracy | (Koh et al., 2020) | Labeled data | Whole Input | Vision |
| $\sim$AUC | (Zimmermann et al., 2023) | Crowdsourced | Whole Input | Vision |
| Inverse AUC | (Bykov et al., 2023) (Kopf et al., 2024) | Labeled data Generative | Whole Input Whole Input | Vision Vision |
| Correlation(T&R) Correlation | (Bills et al., 2023) (Oikarinen & Weng, 2024) | Model Model | Per-token Whole Input | Language Vision |
| Spearman Correlation(T&R) | (Bricken et al., 2023; Templeton et al., 2024) | Model | Per-token | Language |
| $\sim$WPMI | (Oikarinen & Weng, 2023) | Model | Whole Input | Vision |
| MAD | (Kopf et al., 2024) | Generative | Whole Input | Vision |
| $\sim$MAD | (Shaham et al., 2024) (Singh et al., 2023) | Generative Generative | Whole Input Whole Input | Vision Language |

*Table 1.* Summary of **evaluations** used in existing work. $\sim$ indicates using a metric with small differences from our definition, while T&R indicates use of biased *top-and-random* sampling to evaluate the metric with fewer samples. See Table B.1 for an extended versione.

### 3.2. Unifying Neuron Explanation Evaluation: *NeuronEval*

As we have shown in the previous three examples (Eq (4), (5) and (6)), evaluations that look extremely different on the surface, can actually be written as simple functions of the neuron activation vector $a_k$ and concept activation vector $c_t$. This means the explanation score $s_M$ can be written as a function $s_M(a_k, c_t)$ of $a_k$ and $c_t$, that is, $\mathcal{E}(\mathcal{D}, f_k^{0:l}, t) = s_M(a_k, c_t)$, where $M$ is the name of the specific metric chosen. For convenience, we call this the *NeuronEval* Framework, showcased in Figure 1. Going further, we find that, as we show in Table 1, almost all existing methods in the literature for **evaluation** $\mathcal{E}$ can be formalized under this framework.

Notably, *NeuronEval* is a general framework – as shown in Table 1, it includes not only the standard statistical metrics (e.g. Recall, F1-score, Correlation), but also other metrics such as Mean Activation Difference (MAD), which compares the average neuron activation on inputs where the concept is present vs those where concept is missing:

$$s_{MAD} = \frac{B(c_t) \cdot a_k}{||B(c_t)||_1} - \frac{(\mathbf{1} - B(c_t)) \cdot a_k}{||\mathbf{1} - B(c_t)||_1}. \quad (7)$$

Other standard metrics include Precision (Srinivas et al., 2025), which can be intuitively understood as measuring

$\mathbb{P}(\text{neuron is active}|\text{concept})$, defined as:

$$s_{\text{Precision}} = \frac{B(a_k) \cdot B(c_t)}{||B(c_t)||_1} \quad (8)$$

Formalizing metrics under the same framework also highlights gaps in existing work, leading us to test standard statistical metrics that were not explored by previous work, such as AUPRC (defined in Appendix A.3).

Overall we have aimed to include all evaluation metrics used by previous works, as well as standard statistical metrics not previously explored, for a total of 18 different metrics and evaluations from 19 studies within the *NeuronEval* framework, which we compare in our experiments (Section 4 and 5), and the framework can easily include more in the future. In Appendix A.3 we define all the other metrics evaluated, such as F1-score, Cosine similarity, AUC and AUPRC, as well additional details on these metrics.

**Existing Work as Special cases of *NeuronEval*.** We summarize how most existing evaluations fit into our *NeuronEval* framework in Table 1 (see Appendix B.2 for discussion on few exceptions). This includes not only evaluations of individual neuron explanations, but also explanations of SAE features (Bricken et al., 2023), CBM neurons (Koh et al., 2020), or linear probes (Gurnee et al., 2023). Writing

diverse evaluations as special cases of the same framework allows us to more clearly think about each component of an evaluation in isolation, and can help develop more effective evaluations by combining the best parts of previous studies. We classify differences in existing evaluations into four main components:

(i) **Evaluation metric** $M$: This is the main focus of our paper, to analyze which evaluation metrics are good choices.

(ii) **Source of Concept Vector** $c_t$**:** There are many choices for the concept vector $c_t$. These include, but are not limited to: labels from a labeled dataset, using a model to create pseudo-labels, using a human evaluator, or generating new inputs and using the prompts as labels.

(iii) **Granularity of activation vectors:** The simplest case is full input level activations, i.e. $|\mathcal{D}| = |a_k| = |c_t| = n$. These can also be more specific, for example pixel-level activations as is the case in Network Dissection (Bau et al., 2020), or token level as is often the case for language model explanations.

(iv) **Probing dataset** $\mathcal{D}$**:** Which inputs are used for the evaluation. Typical choices include the training/validation data of the model, a special labeled dataset designed for probing, or different datasets for different concepts. Importantly, the dataset used for evaluation should be disjoint from the dataset used for explanation generation to avoid overfitting.

Importantly, regardless of the answer to these questions, or the application domain (e.g. vision or language), each evaluation fits under the same mathematical formalism of *NeuronEval*.

# 4. Meta-Evaluation #1: Sanity Checks for Evaluation Metrics

Now that we have organized previous evaluations under the *NeuronEval* framework, we can isolate and perform *meta-evaluation* on one particular component of the evaluation: the choice of metric $M$. As seen in Table 1, existing work uses a wide variety of metrics, often without justification. In order to understand which evaluation metrics are reliable, we propose two sanity checks: Missing Labels Test and Extra Labels Test which we describe in detail in this section. These tests are inspired by (Adebayo et al., 2018), and passing them is intended to be a necessary (not sufficient) condition for a reliable evaluation metric.

## 4.1. A Motivating Example

We start by analyzing a simple failure case of the precision and recall metrics. Let the probing dataset contain 6 im-

ages of animals $x_i$ ordered as {dog, cat, dog, bear, monkey, flamingo}. Suppose we have a "pets" neuron that only activates on pets (i.e. dogs or cats), then we can compute the neuron activation vector $a_k$ as $B(a_k) = [1, 1, 1, 0, 0, 0]^\top$. We can also compute the concept activation vector $c_t$ for different concepts $t$ based on whether the input image $x_i$ contain the concepts, giving us: $c_{\text{dog}} = [1, 0, 1, 0, 0, 0]^\top$, $c_{\text{cat}} = [0, 1, 0, 0, 0, 0]^\top$, $c_{\text{pet}} = [1, 1, 1, 0, 0, 0]^\top$, $c_{\text{animal}} = [1, 1, 1, 1, 1, 1]^\top$. We can then calculate the neuron explanation score $s_M$ of this "pets" neuron with different metrics $M$ in the following table:

| Explanation $t$ | $s_{\text{Recall}}$ | $s_{\text{Precision}}$ | $s_{\text{IoU}}$ |
|---|---|---|---|
| *dog* | 0.67 | 1 | 0.67 |
| *cat* | 0.33 | 1 | 0.33 |
| *pet* (ground truth) | 1 | 1 | 1 |
| *animal* | 1 | 0.5 | 0.5 |

*Table 2.* A "pet" neuron: different metrics $M$ gives different evaluation scores $s_M$ on explanations $t$. For the metrics $M=\{$Recall, Precision, IoU$\}$, 1 is the perfect score, and 0 is the worst score.

When comparing different evaluation metrics, we can see that *Recall* cannot distinguish between the correct concept (pet) and a concept that is too generic (animal), as $s_{\text{recall}}$ gives perfect score for both explanations. On the other hand, *Precision* favors concepts that are too specific (dog, cat), as $s_{\text{precision}} = 1$ for 'dog', 'cat' and 'pet'. In contrast, IoU can unambiguously determine the correct concept 'pet', as $s_{\text{IoU}} = 1$ only for 'pet'. See Figure B.1 for illustration.

## 4.2. Sanity Test Definitions

Inspired by this example, we propose two general tests to measure whether a certain metric is too biased towards specific or generic concepts. These tests are motivated by the following idea: *For a reliable metric, a correct description for a neuron should get the highest score, while its subset (too specific) or superset (too general) should score lower.*

### (I): Missing Labels Test

The missing labels test is a generalized test to measure whether a metric can differentiate between the correct concept $c_t$ and a concept $c_t^-$ that is too specific (a random subset of the correct concept). In particular we generate $c_t^-$ by randomly replacing half of the elements of $c_t$ with 0. That is we remove half of the concept labels for concept $t$. $\mathbb{E}[||c_t^-||_1] = ||c_t||_1/2$:

$$[c_t^-]_i = \begin{cases} [c_t]_i & \text{with probability } 0.5 \\ 0 & \text{with probability } 0.5 \end{cases} \quad (9)$$

We then measure the score difference $\Delta s$, i.e. how much does the evaluation score change when replacing the original concept vector $c_{t_k}$ with a concept that is too specific, $c_t^-$:

$$\Delta s(k) = \mathbb{E}_{c_{t_k}^\pm} [s_{\text{M}}(a_k, c_{t_k}^\pm) - s_{\text{M}}(a_k, c_{t_k})] \quad (10)$$

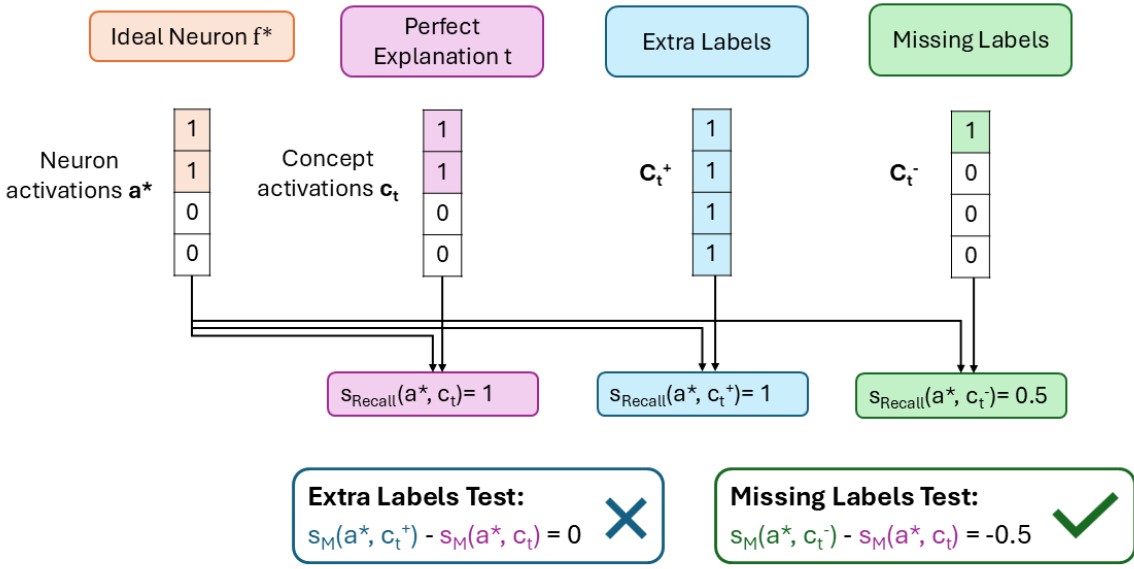

Figure 2. Overview of our theoretical Missing and Extra Labels Tests. We can see the Recall metric fails the extra labels test as it cannot differentiate between the perfect explanation and an explanation with extra labels. On the other hand Recall clearly passes the missing labels test as $-0.5 \ll -\epsilon$ = -0.001

For the sanity test, it is not important how much the modified labels decrease the score. We simply want it to decrease score by a non-zero amount on all neurons, which leads us to define our main metric: *Decrease Acc*.

$$\text{Decrease Acc} = \frac{1}{|K|} \sum_{k \in K} \mathbb{1}[\Delta s(k) < -\epsilon] \quad (11)$$

Here $K$ is the set of neurons looked at and $t_k$ is the best/correct concept for neuron $k$. Note that we normalized the scores such that maximum of $M$ is 1 and minimum value is 0 to allow for equal comparison between metrics. We set $\epsilon = 0.001$ in our experiments to require a small but noticeable change in scores. We study the sensitivity of these tests to the choice of $\epsilon$ further in Appendix E.4. The assumption behind this test is that if concept $t_k$ is a good description for neuron $k$, a random subset of it should be a worse description.

**(II): Extra Labels Test**

The extra labels test is essentially the opposite of missing labels test, where we test whether a metric can differentiate between a correct concept and a concept that is too generic i.e. a random superset of the concept. In particular, we create $c_t^+$ by randomly doubling the size of $c_t$, i.e. $\mathbb{E}[(||c_t^+||_1)] = 2||c_t||_1$. That is:

$$[c_t^+]_i = \begin{cases} 1 & \text{if } [c_t]_i = 1, \text{ else with probability } \frac{||c_t||_1}{n - ||c_t||_1} \\ 0 & \text{otherwise} \end{cases}$$

$$(12)$$

where $n$ is the length of vector $c_t$. Similar to the Missing labels test, to pass the Extra Labels test, a metric should have a high Decrease Acc as defined in Eq. (11). If concept $t$ is a good description for neuron $k$, giving additional positive concept labels to random inputs should decrease its similarity score. For simplicity, we only apply these tests with ground truth labels as concept source where $c_t$ is binary.

### 4.3. Test Setup

We perform two versions of these tests, **Experimental** on real neurons across diverse settings, and **Theoretical** on *ideal* neurons described below.

**Experimental:** We experimentally evaluated these metrics on neurons from 8 different settings across vision and language models, covering final layer neurons, hidden layer neurons, CBM neurons and linear probe outputs. We evaluated vision models across 3 datasets: Imagenet, Places365 and CUB200, while language models were evaluated on a subset of OpenWebText(Gokaslan et al., 2019). See Appendix F for detailed description of the evaluation setting and results on individual datasets. We report the Averaged results in Table 3.

**Theoretical:** In addition, we perform theoretical analysis on the Missing/Extra labels test for hypothetical neurons whose activations perfectly match a concept, which we discuss in detail in Appendix C. See Figure 2 for an overview. In this setting, the missing labels test measures whether a metric can differentiate a concept that perfectly matches neuron

| Meta-Evaluation #1 | (I) Missing Labels Test | | (II) Extra Labels Test | | Pass |
|---|---|---|---|---|---|
| | Experimental | Theoretical | Experimental | Theoretical | |
| **Recall** | 98.66% | 100.00% | 0.00% | 0.00% | ✗ |
| **Precision** | 45.73% | 0.00% | 99.81% | 100.00% | ✗ |
| **F1-score** | 93.68% | 100.00% | 99.82% | 100.00% | ✓ |
| **IoU** | 93.62% | 100.00% | 99.81% | 100.00% | ✓ |
| **Accuracy** | 23.79% | 60.00% | 70.37% | 69.68% | ✗ |
| **Balanced Accuracy** | 98.65% | 100.00% | 53.67% | 60.00% | ✗ |
| **Inverse Balanced Accuracy** | 64.18% | 60.00% | 99.50% | 100.00% | ✗ |
| **AUC** | 94.96% | 100.00% | 59.18% | 60.00% | ✗ |
| **Inverse AUC** | 52.81% | 60.00% | 99.99% | 100.00% | ✗ |
| **Correlation** | 99.41% | 100.00% | 99.92% | 100.00% | ✓ |
| **Correlation (T&R)** | 87.83% | 100.00% | 60.26% | 43.64% | ✗ |
| **Spearman Correlation** | 64.05% | 67.20% | 49.21% | 44.08% | ✗ |
| **Spearman Correlation (T&R)** | 80.04% | 100.00% | 59.81% | 19.68% | ✗ |
| **Cosine** | 99.45% | 100.00% | 99.26% | 100.00% | ✓ |
| **WPMI** | 95.89% | 100.00% | 58.84% | 100.00% | ✗ |
| **MAD** | 59.81% | 60.00% | 99.34% | 100.00% | ✗ |
| **AUPRC** | 95.61% | 100.00% | 99.46% | 100.00% | ✓ |
| **Inverse AUPRC** | 99.15 % | 100.00% | 95.58% | 89.54% | ✗ |

*Table 3.* Averaged experimental and theoretical results of our missing labels and extra labels test. Experimental results are averaged across settings like vision and language domain, while theoretical results are averaged over different neuron activation frequencies. (T&R) indicates the use of top-and-random sampling. We can see most evaluation metrics fail at least one of the tests.

activations $c_t$ (Recall=1, Precision=1) from a concept $c_t^-$ (with Recall=0.5 and Precision=1). Similarly, the Extra Labels test measures differentiation between the perfect concept and $c_t^+$ (with Recall=1 and Precision=0.5). We simulate neurons with many different activation frequencies, and averaged results are reported in Table 3.

Finally, we analytically studied the expected change in scores for different binary metrics under Missing and Extra labels test in Appendix D. Our analytical results are consistent with the empirical findings in Table 3 and Fig. 3.

### 4.4. Results and Discussion

**Results:** In Table 3 we report the averaged results of this test across these two sets of neurons for all different evaluation metrics. For simplicity, we say a metric passes if its Decrease Acc $> 90\%$ for all tests. Failing methods are marked in red color. Overall, our theoretical results closely match our empirical observations. Based on test results, we can group the metrics as follows:

- **Fail Both:** *Accuracy* and *Spearman Correlation* perform poorly in both tests as their score is largely determined by the majority of inputs that neither activate the neuron nor have the concept.

- **Fail Missing Labels Test:** *Precision*, *Inverse Balanced Accuracy*, *Inverse AUC* and *MAD* are biased towards too specific concepts.

- **Fail Extra Labels Test:** *Recall*, *Balanced Accuracy*,

*AUC* and *Correlation(T&R)* are biased towards concepts that are too generic.

- **Borderline:** Our results are inconclusive for the following two metrics: *WPMI* which passes the tests on certain hyperparameter choices but not on others, and *Inverse AUPRC* fails the theoretical test only on perfectly balanced data but passes the empirical tests.

- **Passing:** *F1-score*, *IoU*, *Correlation*, *Cosine similarity* and *AUPRC* perform well and pass the tests.

Importantly, we think any metric failing these tests should not be relied on by itself, and encourage the use of passing metrics. We also perform an Ablation in Appendix E.1 showing that our findings are not sensitive to specific parameter choices of the test.

**Concept Imbalance Explains Test Failures:** In our theoretical experiments, we discover that the main reason for a metric failing one of these tests is poor handling of concept imbalance. Figure 3 shows an example of this phenomenon. On balanced neurons/concepts that activate on more than 10% of the inputs, all metrics except for Recall pass the Extra Labels Test (large $|\Delta s|$). However, on neurons that activate less frequently, the failing metrics such as Accuracy and AUC can no longer distinguish between the perfect concept and an overly generic concept, and the score difference approaches 0. Therefore, failing the Missing/Extra Labels test can in most cases be attributed to the metrics being unable to handle imbalanced data. In this light, our

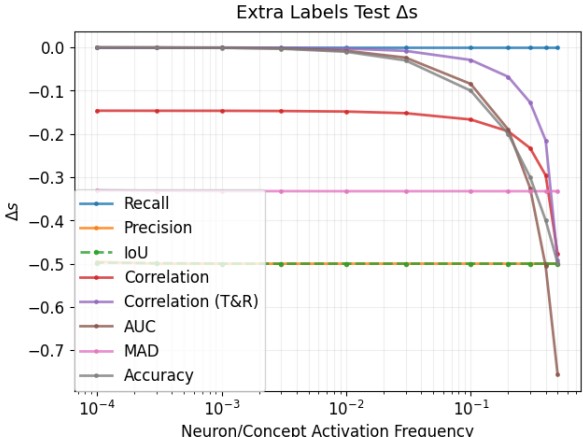

*Figure 3.* The effect of Concept/Neuron Activation frequency on the score difference in Theoretical Extra Labels test. We can see that all the metrics that fail the Extra Labels test approach 0 score difference on neurons/concepts that rarely activate, i.e. where the concept labels are imbalanced. The passing metrics maintain a non-zero score difference regardless of how rare the concept is.

results align with previous statistical knowledge, and metrics such as Accuracy and AUC which are not great for imbalanced datasets fail the tests, while metrics designed for imbalanced data such as F1-score and AUPRC pass the tests. Since real units in neural network often activate on a wide range of frequencies, and are sometimes encouraged to activate very rarely (i.e. Sparse Autoencoders), it is essential to use evaluation metrics that can handle imbalanced data.

**Top-and-random sampling:** Following (Bills et al., 2023; Bricken et al., 2023), we also tested a version of Correlation and Spearman Correlation that uses top-and-random sampling (T&R), where 50% of the evaluated samples are highly activating inputs only and the remaining 50% sampled uniformly at random. For our T&R experiments we sampled 25 inputs from the top 0.2% highest activating inputs and 25 random inputs. While *Correlation* passes both tests with uniform sampling, top-and-random sampling makes it fail the extra labels test. This is not surprising, as greatly oversampling highly activating inputs makes the metric more similar to Recall that only evaluates on highly activating inputs. This also explains why (Huang et al., 2023) found explanations from (Bills et al., 2023) with very high Correlation(T&R) scores to have relatively low F1-scores.

$c_t$ **from Generative models:** We discover that evaluations based on generating new inputs with generative models are limited to running evaluations that fail the missing labels test and are biased towards more specific concepts. This is because $c_t$ based on generative model prompts is inherently incomplete or *missing labels*, and as such should not be solely relied on. See Appendix B.1 for further discussion on this situation and potential remedies.

# 5. Meta Evaluation #2: Performance on Neurons with Known Concepts

In this section we perform an additional comparison between evaluation metrics by empirically comparing how well they perform on neurons where we know their *ground truth* function, such as neurons in final (classification) layer of a model.

## 5.1. Method: AUPRC between scores of correct and incorrect explanations

The idea of this test is to directly measure how good specific evaluation metrics are. To do this, we have a concept set $C$ that includes the correct explanation for each neuron $k \in K$, as well as some incorrect explanations. $K$ is the set of neurons evaluated such as all neurons in a layer. We then score all $|K||C|$ (neuron, explanation) pairs and measure whether the metric consistently assigns higher scores to pairs with correct explanations over incorrect ones.

In particular, we measure the AUPRC (area under precision-recall curve) since this is a very imbalanced task with most (neuron, explanation) pairs being incorrect. Mathematically, we can write the predictions $\hat{Y}_M$ as: $\hat{Y}_M = \{s_M(a_k, c_t) \quad \forall \quad k \in K, t \in C\}$, while the labels $Y$ are $Y = \{\mathbb{1}[t = t_k^*]) \quad \forall \quad k \in K, t \in C\}$, where $t_k^*$ is the correct explanation for that neuron. Our meta-AUPRC evaluation can then be defined as:

$$\text{meta-AUPRC(M)} = \text{AUPRC}(\hat{Y}_M, Y) \qquad (13)$$

A metric reaches perfect meta-AUPRC of 1 if all correct (neuron, explanation) pairs get a higher score than all incorrect pairs, while random guess performance is $\sim 0$.

**Experimental Setup.** Similar to section 4, we ran these test on 10 different setups, consisting of 5 separate models, 4 datasets, vision and language domains and both ground truth labels and pseudo-labels as concept source for $c_t$. See Appendix F for detailed description of experimental setup and details on individual models.

For all experiments we split a random 5% of the neurons into validation set. For metrics that require hyperparameters such as $\alpha$, we use the hyperparameters that performed the best in terms of Meta-AUPRC on the validation split for each setting. We then report performance on the remaining 95% of neurons. In Table 4, we report the average scores and average ranks (i.e. the best metric for each setup gets 1, the worst gets 17) of the metrics across the four setups for both tasks. See Appendix F for detailed breakdown of results.

We can see that in general the metrics that passed our tests in Section 4 perform better than those that didn't, with the top performance going to Correlation, followed by Cosine simi-

| Method | Avg. AUPRC | Avg. Rank |
|---|---|---|
| Recall | 0.6722 | 11.30 |
| Precision | 0.8039 | 7.90 |
| F1-score/IoU | 0.8140 | 6.70 |
| Accuracy | 0.7215 | 10.80 |
| Balanced Accuracy | 0.7979 | 7.30 |
| Inverse Balanced Acc. | 0.8087 | 7.10 |
| AUC | 0.7652 | 11.00 |
| Inverse AUC | 0.7569 | 10.90 |
| Correlation | **0.8765** | **1.60** |
| Correlation(T&R) | 0.6606 | 10.70 |
| Spearman Correlation | 0.0853 | 16.20 |
| Spearman Correlation(T&R) | 0.3418 | 15.40 |
| Cosine | 0.8666 | 2.30 |
| WPMI | 0.7999 | 7.30 |
| MAD | 0.6952 | 8.80 |
| AUPRC | 0.8406 | 3.90 |
| Inverse AUPRC | 0.6904 | 9.30 |

*Table 4.* Comparison of different evaluation metrics on neurons with known concepts, averaged across 8 settings. Lower rank means better performance. Best performing metric in **bold**, and second best underlined.

larity then AUPRC and F1-score/IoU. [3] Overall we notice continuous metrics perform somewhat better than binary ones. This is likely because binarizing neuron activations loses valuable information, and it is hard to find one binarization threshold $\alpha$ that works well for all neurons. Despite good performance here, the *Cosine* metric is very sensitive to the average activation of a neuron as we show in E.2, which raises some concerns regarding relying on it.

## 6. Discussion

### 6.1. Relevance of our tests to Realistic Failure modes

While our sanity test definition in Section 4 is relatively abstract, we think that most real world explanation failures are closely connected to one of two failure modes, and that these failure modes are directly captured by our tests:

- **FM-A) Explanation too generic:** This means the explanation concept is a superset of the "true" neuron concept, e.g. describing a neuron as "animals" when it only activates on dogs. Our Extra Labels Test captures whether a metric can detect this failure mode.

- **FM-B) Explanation too specific:** Explanation concept is a subset of real neuron activations, i.e. describing a neuron as "black cat" when it activates on all cats. Our Missing Labels Test captures whether a metric can detect this failure mode.

---

[3] Note we combined F1-score and IoU into same element in this table as they always return the same ordering between pairs leading to identical AUPRC, see Appendix A.4 for proof.

Since our idea of a "concept" is very general, it can include any text-based description. This means a single concept could be highly specific (e.g. "flying bird"), or a composition of simpler concepts (e.g. "water OR river"). Below we show how common explanation failures relate to our tests.

- **Polysemanticity:** A popular model of polysemanticity is to model neuron activations as an OR of different concepts. If the explanation only captures one of these concepts, this means the explanation is too specific (FM-B).

- **Context specific activations:** A context specific neuron activation such as *flying bird* means the neuron's "true" concept is a subset of the non-context specific concept i.e. *bird*. If the explanation is not context-specific, the explanation is too generic (FM-A).

- **Non activating inputs still contain the concept:** This is a common explanation failure case also known as spurious correlations, where all the highly activating inputs share a concept, but not all inputs with this concept make the neuron activate. This means that the explanation concept is too generic i.e. a superset of the "true" concept (FM-A).

Finally, in Appendix E.3 we conducted a test using real semantic superclasses of the concept as $c^+$ instead of our random superset, and found that the results are essentially identical, highlighting the relevance of our tests to real world settings.

### 6.2. Conclusions

In this paper, we have created the *NeuronEval* framework for unifying different evaluations under the same mathematical formalism, clarifying the definitions of around evaluating unit explanations. We have also proposed new sanity tests which lead to our main finding: **Neuron Explanation Evaluations should use metrics that pass the Missing and Extra Labels Tests** – We identified Correlation, Cosine, AUPRC, F1-score and IoU as solid passing metrics in our study of 18 different metrics. Popular evaluation metrics used in the literature such as Recall and AUC as well as using top-and-random sampling do not pass these tests, and should not be relied on by themselves. With evaluation on neurons with ground truth explanation available (e.g. final layer neuron), we identified that the top-performing metrics overall are Correlation, Cosine and AUPRC. The final choice of metric may be further affected by considerations such as labeling cost discussed in B.2 or other failure modes such as Cosine's sensitivity to average activation showcased in E.2.

## Impact Statement

Our paper is focused on rigorous evaluation of neural network interpretability results, and as such we believe it will have a largely positive societal impact. Being able to understand/see inside models is likely to be very useful in deploying models in a reliable and safe manner. In addition, rigorously evaluating their interpretations is very important so we can avoid relying on unfaithful explanations or interpretability illusions and can place appropriate trust in these models.

## Acknowledgements

The authors are partially supported by National Science Foundation under Grant No. 2107189, 2313105, 2430539, Hellman Fellowship, Intel Rising Star Faculty Award, Nvidia Academic Grant Program as well as the U.S. National Science Foundation Institute for Learning-enabled Optimization at Scale (TILOS; NSF CCF-2112665). The authors would like to thank anonymous reviewers for valuable feedback to improve the manuscript.

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

# Appendix

# A. Metric Definition & Details

## A.1. Binarization

Many popular metrics in the literature require inputs to be binary such as (Bau et al., 2017; Huang et al., 2023). Since neuron activations and concept activations from some sources are continuous, we need to binarize these vectors. We denote this with the binarization function $B : \mathbb{R}^n \to \{0,1\}^n$.

Typically for neuron activation $a_k$, following previous work we use $B = \text{top}_\alpha$, where we take top $\alpha$ fraction of activations to be 1, and others to be 0. We formalize this as $\text{top}_\alpha(z)$:

$$[\text{top}_\alpha(z)]_i = \begin{cases} 1 & \text{if } z_i \geq b_\alpha; \\ 0 & \text{otherwise} \end{cases} \tag{A.1}$$

where $b_\alpha$ satisfies $\sum_{i=1}^n \frac{\mathbb{1}[z_i \geq b_\alpha]}{n} = \alpha$, and $z \in \mathbb{R}^n$. For example, if $\alpha = 0.05$, then $\text{top}_\alpha$ has 1's for inputs with activations in top-5%, and 0 for others. Note $\alpha$ is a hyperparameter needed for all binary similarity functions. We typically select $\alpha$ independently for each metric by finding the value that performs the best on a small validation split of neurons. For concept vectors $c_t$, we usually binarize simply by rounding, denoted as $B = r$, where:

$$[r(z)]_i = \begin{cases} 1 & \text{if } z_i \geq 0.5 \\ 0 & \text{if } z_i < 0.5 \end{cases} \tag{A.2}$$

## A.2. Simulation vs Classification Framing

**Simulation:** In our definitions in section 3 we use the neuron activation $a_k$ as the *ground truth*, and concept presence $c_t$ as our prediction. This corresponds to framing the evaluation as *simulation*, i.e. trying to predict neuron activation based on concept value. Given this, we can express the Binary Classification Confusion Matrix elements (True Positive (TP), False Positive (FP), False Negative (FN) and True Negative (TN)) in terms of the vectors $a_k$ and $c_t$ as:

$$\text{TP} = B(a_k) \cdot B(c_t), \text{ FP} = \overline{B(a_k)} \cdot B(c_t),$$
$$\text{FN} = B(a_k) \cdot \overline{B(c_t)}, \text{ TN} = \overline{B(a_k)} \cdot \overline{B(c_t)},$$

where $\overline{B(\cdot)}$ represents element-wise NOT operation on the binary vector (equivalent to $\mathbf{1} - B(\cdot)$ where $\mathbf{1}$ is a vector of all 1's.)

This corresponds to seeing the explanation from a **simulation** point of view, i.e. our goal is to predict how the neuron activates, based on the neuron's explanation and current inputs. This gives us binary classification metric definitions that are aligned with those of (Huang et al., 2023).

**Classification:** However this is an arbitrary choice, and we could just as well define $c_t$ as the *ground truth* and $a_k$ as the prediction. This corresponds to a classification view, where our goal is to use neuron $k$ as a classifier for concept $t$. In terms of metrics, this doesn't change the definitions for True Positive (TP) or True Negative (TN), but it switches the places of False Positive and False Negative, i.e.

$$\text{TP}_{cls} = B(a_k) \cdot B(c_t) = \text{TP}_{sim}$$
$$\text{FP}_{cls} = B(a_k) \cdot \overline{B(c_t)} = \text{FN}_{sim}$$
$$\text{FN}_{cls} = \overline{B(a_k)} \cdot B(c_t) = \text{FP}_{sim}$$
$$\text{TN}_{cls} = \overline{B(a_k)} \cdot \overline{B(c_t)} = \text{TN}_{sim}$$

This change in framing also affects for metrics which are not symmetric in terms of False Positives and False Negatives. For example, Recall(simulation) = Precision(classification), and Precision(simulation) = Recall(classification).

$$\text{Recall(simulation)} = \frac{\text{TP}_{sim}}{\text{TP}_{sim} + \text{FN}_{sim}} = \frac{\text{TP}_{cls}}{\text{TP}_{cls} + \text{FP}_{cls}} = \text{Precision(classification)}$$

The other binary metric that is sensitive to this framing is balanced accuracy. The metric we call Balanced Accuracy in Appendix A.3 corresponds to the simulation version of Balanced Accuracy, so for completeness sake we also include Inverse Balanced Accuracy (Appendix A.3), which is the classification version.

Of the non-binary metrics, AUC and AUPRC are sensitive to this framing, and the standard versions AUC and AUPRC we define in Appendix A.3 are AUC(simulation) and AUPRC(simulation), while we also include classification versions of these metrics labeled as Inverse AUC/AUPRC.

We note that some related works use the simulation framing, while others such as (Zhou et al., 2015) use the classification framing, which leads to conflicting definitions of Precision and other metrics.

### A.3. Additional Metric Definitions

1. **Accuracy:** Standard binary accuracy. Percentage of time the neuron activation matches concept activation.

$$s_{\text{Acc}} = \frac{TP + TN}{TP + FP + FN + TN} = \frac{B(a_k) \cdot B(c_t) + (\mathbf{1} - B(a_k)) \cdot (\mathbf{1} - B(c_t))}{n} \tag{A.3}$$

2. **F1-score:** F1-score is the harmonic mean of precision and recall, and can be expressed as:

$$s_{F1} = \frac{2 \cdot B(a_k) \cdot B(c_t)}{||B(a_k)||_1 + ||B(c_t)||_1} \tag{A.4}$$

3. **Balanced Accuracy:** A version of accuracy designed for imbalanced datasets that averages the accuracy on positive and negative inputs.

$$s_{BA} = \frac{B(a_k) \cdot B(c_t)}{2||B(a_k)||_1} + \frac{(\mathbf{1} - B(a_k)) \cdot (\mathbf{1} - B(c_t))}{2||(\mathbf{1} - B(a_k))||_1} \tag{A.5}$$

4. **Inverse Balanced Accuracy:** Balanced accuracy but we consider $a_k$ to be the prediction and $c_t$ to be the ground truth.

$$s_{IBA} = \frac{B(a_k) \cdot B(c_t)}{2||B(c_t)||_1} + \frac{(\mathbf{1} - B(a_k)) \cdot (\mathbf{1} - B(c_t))}{2||(\mathbf{1} - B(c_t))||_1} \tag{A.6}$$

5. **AUC:** Area under ROC curve also known as AUROC, can be efficiently calculated as:

$$s_{\text{AUC}} = \frac{\sum_{i|B(a_k)_i=0} \sum_{j|B(a_k)_j=1} \mathbb{1}[c_{ti} < c_{tj}] + 0.5 \cdot \mathbb{1}[c_{ti} = c_{tj}]}{||B(a_k)||_1||1 - B(a_k)||_1} \tag{A.7}$$

6. **Inverse AUC:** Area under receiving-operating-characteristics(AUROC) curve, where we consider $a_k$ to be the prediction and $c_t$ to be the ground truth(classification framing).

$$s_{IAUC} = \frac{\sum_{i|B(c_t)_i=0} \sum_{j|B(c_t)_j=1} \mathbb{1}[a_{ki} < a_{kj}] + 0.5 \cdot \mathbb{1}[a_{ki} = a_{kj}]}{||B(c_t)||_1||1 - B(c_t)||_1} \tag{A.8}$$

7. **Spearman Correlation:** The Spearman correlation is equivalent to the Pearson Correlation between the ranks of elements.

$$s_{Spear} = \frac{1}{n} \frac{(R(a_k) - \mu(R(a_k))) \cdot (R(c_t) - \mu(R(c_t)))}{\sigma(R(a_k))\sigma(R(c_t))} \tag{A.9}$$

Here $R$ is a function that elementwise returns the ranks of each element.

8. **Cosine similarity:** The standard cosine similarity between two vectors.

$$s_{cos} = \frac{a_k \cdot c_t}{||a_k||_2||c_t||_2} \tag{A.10}$$

9. **WPMI:** Weighted pointwise-mutual information. A version of this objective is used by (Hernandez et al., 2022) and (Oikarinen & Weng, 2023) to generate explanations, and by (Oikarinen & Weng, 2023) to evaluate said explanations.

$$s_{WPMI} = \sum_{i|B(a_k)_i=1} [\log(c_{ti}) - \lambda \log(\mu(c_t))] \qquad (A.11)$$

In the above equations $n$ is the length of $a_k$ and $c_t$, $\mu$ calculates the mean of the vector and $\sigma$ its standard deviation. $\lambda$ is a hyperparameter and $R$ is the rank operator, which transforms each element to its rank, with smallest element becoming 1 and largest $n$.

10. **AUPRC:** Area Under Precision-Recall Curve(AUPRC) is a popular metric for measuring classification performance, in particular for imbalanced data. While we are not aware of a closed form solution, it can be calculated as:

   (a) Calculate precision and recall at each threshold $\tau_i$, where threshold contains distinct values of $c_t$. Recall $R_i = \frac{B(a_k)\mathbf{1}(c_t \geq \tau_i)}{\|B(a_k)\|_1}$, precision $P_i = \frac{B(a_k)\mathbf{1}(c_t \geq \tau_i)}{\|\mathbf{1}(c_t \geq \tau_i)\|_1}$.
   (b) Calculate area under precision-recall curve using numerical integral:

$$s_{AUPRC} = \sum_n (R_i - R_{i-1})P_i$$

   AUPRC outputs values in $[0, 1]$ range.

11. **Inverse AUPRC:** Same as AUPRC, but with a differenct framing so we flip $c_t$ and $a_k$ in the calculations.

   See Tables A.1 and A.2 for additional details on our metrics.

### A.4. Equivalences

During our analysis we also notice that certain separate metrics are equivalent or very similar to each other.

**Correlation and Cosine similarity:** Calculating correlation between two vectors is equal to normalizing each vector to have mean 0 and then taking their cosine similarity, as shown below:

Let $\hat{x} = x - \mu(x)$ for any vector $x$. Then

$$\text{Cosine}(\hat{a_k}, \hat{c_t}) = \frac{\hat{a_k} \cdot \hat{c_t}}{||\hat{a_k}||_2 ||\hat{c_t}||_2} = \frac{(a_k - \mu(a_k)) \cdot (c_t - \mu(c_t))}{\sqrt{n}\sigma(a_k)\sqrt{n}\sigma(c_t)} =$$

$$\frac{1}{n} \frac{(a_k - \mu(a_k)) \cdot (c_t - \mu(c_t))}{\sigma(a_k)\sigma(c_t)} = \text{Correlation}(a_k, c_t) \qquad (A.12)$$

This explains why the two perform very similarly in our evaluations.

**IoU and F1-score:** Below we show that IoU and F1-score are very closely related. In fact, F1-score can be written as a monotonously increasing function of IoU. This means that for any vectors $x_1, y_1, x_2, y_2$, $\text{IoU}(x_1, y_1) < \text{IoU}(x_2, y_2) \rightarrow \text{F1}(x_1, y_1) < \text{F1}(x_2, y_2)$, so for the purposes of comparing similarites they behave identically, and the choice of which one to use doesn't matter. As their performance was exactly the same in all tasks, we report them in the same row in Table 4.

Intersection over Union (IoU) also known as Jaccard index is defined as

$$\text{IoU} = \frac{TP}{TP + FP + FN} \qquad (A.13)$$

while F1-score also known as Dice-score is defined as:

$$\text{F1} = \frac{2TP}{2TP + FP + FN} \qquad (A.14)$$

Now

$$F1 = \frac{2TP}{2TP + FP + FN} = \frac{2TP}{TP + FP + FN} \cdot \frac{TP + FP + FN}{2TP + FP + FN} \tag{A.15}$$

$$= 2\text{IoU} \cdot \left(\frac{2TP + FP + FN}{TP + FP + FN}\right)^{-1} = \frac{2 \cdot \text{IoU}}{\text{IoU} + 1} \tag{A.16}$$

Which is monotonously increasing for $0 \leq \text{IoU} \leq 1$. So using either metric gives the same comparative results.

| Metric | Definition | Range |
|---|---|---|
| Recall | TP / (TP + FN) | $[0, 1]$ |
| Precision | TP / (TP + FP) | $[0, 1]$ |
| F1-score | 2TP / (2TP + FP + FN) | $[0, 1]$ |
| IoU | TP / (TP + FP + FN) | $[0, 1]$ |
| Accuracy | (TP + TN) / (TP + FP + TN + FN) | $[0, 1]$ |
| Balanced accuracy | [TP / (TP + FN) + TN / (TN + FP)] / 2 | $[0, 1]$ |
| Inverse balanced accuracy (classification version of balanced accuracy (see App. A.2)) | [TP / (TP + FP) + TN / (TN + FN)] | $[0, 1]$ |

*Table A.1.* Definition of commonly-used binary classification metrics. Here, TP, FP, TN, FN refer to true positive, false positive, true negative and false negative, respectively.

| Metric | Definition | Range |
|---|---|---|
| AUC (swap $x$ and $y$ to get inverse AUC) | $\dfrac{\sum_{y_i=1} \sum_{y_j=0} [\mathbf{1}\{x_i > x_j\} + 0.5 * \mathbf{1}\{x_i = x_j\}]}{|y=1||y=0|}$ | $[0, 1]$ |
| Correlation | $\dfrac{\sum (x_i - \bar{x})(y_i - \bar{y})}{\sqrt{\sum (x_i - \bar{x})^2 \sum (y_i - \bar{y})^2}}$ | $[-1, 1]$ |
| Spearman correlation | Replace $x, y$ to corresponding rank $R(x), R(y)$ in correlation | $[-1, 1]$ |
| Cosine | $\dfrac{\sum x_i y_i}{\|x\|_2 \|y\|_2}$ | $[-1, 1]$ |
| WPMI | $\log p(x \mid y) - \lambda \log(p(x))$ | $(-\infty, \infty)$ |
| MAD | $\dfrac{\sum_{y_i=1} x_i}{|y_i = 1|} - \dfrac{\sum_{y_i=0} x_i}{|y_i = 0|}$ | $(-\infty, \infty)$ |

*Table A.2.* Definition of other commonly-used metrics. Here, $x, y \in \mathbb{R}^N$ are two real vectors, $\bar{x}, \bar{y}$ refer to the mean of $x, y$.

# B. Discussion

## B.1. Generative Model based Evaluation and missing labels test

$c_t$ **from Generative Models:** Evaluation methods that use generative models to generate new data (Kopf et al., 2024; Shaham et al., 2024; Singh et al., 2023) effectively define the concept vector $[c_t]_i = 1$ if $x_i$ was generated with prompt $t$, and 0 otherwise. However, this concept vector is incomplete, as some of "negative" examples that were not generated with that prompt may still contain the concept. In effect, the concept vector $c_t$ is actually naturally missing labels similar to $c_t^-$ from our missing labels test. This is because the generated inputs serve as positive labels for $c_t$, but the negative inputs are often taken to just be all inputs from a dataset, even though some of actually do have the concept $t$. So the $c_t$ we get from generative models is actually randomly missing a potentially large portion of the labels. Because of this, most evaluation methods using generative $c_t$ in Table 1 use methods that fail the missing labels test such as Inverse AUC and MAD. In fact, this is desirable with this concept source, as the missing labels in $c_t$ do not affect the explanation score with these metrics. However, these metrics alone should not be relied for evaluation.

**Potential Solutions and Future Directions:** The best way to use generative models in evaluation might be to combine them with another evaluation that doesn't fail the missing labels test, as is done by (Huang et al., 2023) who measure Precision using generated data, and measure Recall on existing data with model based pseudo-labels and combine these results into F1-score. A useful evaluation in this vein could also be using a crowdsourced evaluation to measure Recall, and combining that with generative model based evaluation of Precision. Another option would be to not use the $c_t$ derived from prompt directly, but instead use another model to estimate $c_t$ on all inputs, not just generated ones which would avoid the missing labels problem. Finally an interesting direction would be to look into combinations of other metrics, for example could AUC and Inverse AUC be combined similarly to how Recall and Precision combine into F1-score. We explore these combination metrics further in Appendix E.6.

## B.2. Limitations

**Framework Limitations:**

Not every evaluation of neuron descriptions can fit into our framework. Below we split these into few separate cases and discuss whether each case represents a limitation of the framework or not:

- **Evaluating Multiple Inputs at once:** Our evaluation framework assumes that the presence of a concept is estimated separately for each input. Many human study based evaluations (e.g. (Bau et al., 2017), (Oikarinen & Weng, 2023)) instead evaluate a group of inputs at once, asking questions like "How well does *concept* match this group of images?". However we believe this is simply a less precise/less objective way of asking whether the concept matches each input separately and does not in general represent a significant limitation for the framework.

- **Comparing similarity to "correct" explanation:** Another approach to evaluate neuron descriptions is to compare how close they are to a "correct" description, typically in a text-embedding space. For example, this is the main evaluation used by MILAN (Hernandez et al., 2022), where they generate "correct" explanation by asking Mechanical Turk workers to describe neurons based on their most highly activating inputs. We do not think this a very reliable way to evaluate explanations, because it relies on the assumption that there exists a single "correct" text-based explanation for each neuron (and that we have some way of finding it), and we do not think this is the case for many real neurons because of issues like polysemanticity and non-verbal concepts like specific graphical patterns. For these hard-to-interpret neurons it is better to just measure how well our explanation matches the neuron like the metrics in our framework do.

- **Non-text based explanations:** While we focus on text based concepts $t$ in our paper, the framework works on non-textual concepts just as well, as long as we have some way of generating a concept vector $c_t$ for that concept. For example, the evaluation of (Zimmermann et al., 2023) uses a group of highly activating inputs as the concept, and then asks workers whether a new input is similar to those inputs or not. Despite this difference, it can be described neatly within our framework.

- **Output Based Neuron Explanations:**

  For clarity, it is useful to divide the neural network $f(x)$ into two parts, $f^{0:l}$ and $f^{l+1:L}$, where 0 corresponds to the input layer, $f^{i:j}$ represents the $i$ through $j$'th layers of the neural network $f(x)$, $l$ is the layer of the neuron we are interested in and $L$ is the total number of layers in the network. Then:

$$f(x) = f^{l+1:L}(f_k^{0:l}(x), f_{\neg k}^{0:l}(x)), \tag{B.1}$$

where $f_k^{0:l}(x)$ is the activation of neuron of interest $k$ in layer $l$, while $f_{\neg k}^{0:l}(x)$ is the activations of all the other neurons in layer $l$.

Alternatively to **Input-based** neuron explanations we define in section 2.3, a neuron explanation could instead be **Output-based**, that is an interpretable approximation of the following function:

– **Unit Activation → Output:** $z \to f^{l+1:L}(z, f_{\neg k}^{0:l}(x))$

Where $z$ is a real number (e.g. intervened value). This function describes how changes in the unit activation change the final network output, as has been the focus of a few recent papers on neuron explanations (Gandelsman et al., 2024; Gur-Arieh et al., 2025).

As we can see from the above notations, output and input-based explanations are different problems, and may require different methods to solve and evaluate. In our paper, we focus on evaluating explanations of **Input-based**, i.e. the function $x \to f_k^{0:l}(x)$, as this is more common in existing evaluations and can be applied more generally, for example to explain linear combinations of neurons that don't have a direct effect on the output such as TCAV (Kim et al., 2018) vectors. While evaluating Output-based explanations is also important, this may require different methods and is outside the scope of our current work.

This may be the most significant limitations of our framework, as we believe measuring Output based explanations is equally important, and an ideal neuron explanation should describe both functions(potentially with different explanations). However, improving the evaluation of Input-based explanations is already a significant contribution towards better evaluation practices. In addition, while currently our framework is meant for Input based evaluation only, we believe many of the ideas and metrics we discussed could be useful in evaluating output based explanations. For example, in a generative models we could use the same metrics to measure similarity between unit activation and the presence of a specific concept in the output. However in output-based evaluations there are additional considerations such as measuring difference in outputs when changing the unit activation that may be more important. We believe extending this framework or creating a similar one for output-based evaluations is an important direction for future work.

**Experimental Result Limitations:**

Overall we are quite confident in the generality of our results on the missing/extra labels test (Table 3) as they are consistent across final/hidden layer neurons and different datasets, and we showed theoretically that they are caused by poor metric performance on imbalanced data. Importantly, this theoretical result is independent of the data domain, type of concepts or the type of unit in question.

However, it is good to note that passing these sanity checks does not guarantee that the metric is a good metric, but failing them does indicate a metric should not be relied on. This is similar to sanity checks proposed by (Adebayo et al., 2018), which have been quite influential in the field of saliency maps/input importance estimation.

On the other hand, our comparison results in Table 4 are mostly focused on final layer neurons or other units where we have ground truth available such a concept neurons inside a CBM. While they consistently prefer certain metrics, it is possible that these final layer neurons are systematically different from other units we are interested in such as hidden layer neurons, and these results should not be relied on too strongly.

**Labeling Cost:**

One of the main reasons existing crowd-sourced evaluations only measure *Recall* is it's lower evaluation cost, as we only need to only evaluate $c_t$ on highly activating inputs, i.e. where $B(a_k) = 1$, because the values of $c_t$ on other inputs do not affect the score. In contrast, evaluating most other metrics requires knowledge of the full $c_t$. Similar cost issues are also faced when using an expensive LLM based simulation pipeline to predict $c_t$ as done by (Bills et al., 2023), causing evaluation metrics to often be evaluated on a small subset instead.

Crowd-sourced evaluation of the entire $c_t$ will likely give noisy results as most inputs do not contain the concept. To avoid this, another approach is to oversample highly activating inputs similarly to Top-and-random sampling, though more effective sampling strategies likely exist. This can be done without failing the extra labels test if the proper sampling correction is applied to correct for the bias introduced by the non-uniform sampling. As our paper is focused on the correctness of

different metrics, not the ease of measuring them, and the effective sampling/user study design for non-recall metrics is a rather large and complex topic that does not fit within the scope of the current paper, but it is a promising direction for future work.

An alternative way to reduce labeling costs is to use a combination of metrics, for example evaluating F1-score by combining a crowd-sourced evaluation of Recall with a generative model based evaluation of Precision.

### B.3. Motivating Example for Sanity Checks

Here, we illustrate the motivating example discussed in Section 4.1 in the following Figure. Figure B.1 displays an example neuron that only activates on images containing pets to highlight how similarity scores are calculated. We can see that only measuring Recall gives a perfect score to explanations that are too generic (*animal*), while only measuring precision favors explanations that are too specific(*dog*, *cat*).

**Example neuron activating on pets only**

| | | | | | | |
|---|---|---|---|---|---|---|
| $B(a_k)$ | 1 | 1 | 1 | 0 | 0 | 0 |
| $c_{dog}$ | 1 | 0 | 1 | 0 | 0 | 0 |
| $c_{cat}$ | 0 | 1 | 0 | 0 | 0 | 0 |
| $c_{pet}$ | 1 | 1 | 1 | 0 | 0 | 0 |
| $c_{animal}$ | 1 | 1 | 1 | 1 | 1 | 1 |

**Results of different evaluation metrics:**

| Concept t | $s_{Recall}$ | $s_{Precision}$ | $s_{IoU}$ |
|---|---|---|---|
| Dog | 2/3 = 0.67 | 2/2 = 1 | 2/3 = 0.67 |
| Cat | 1/3 = 0.33 | 1/1 = 1 | 1/3 = 0.33 |
| **Pet** | **3/3 = 1** | **3/3 = 1** | **3/3 = 1** |
| Animal | 3/3 = 1 | 3/6 = 0.5 | 3/6 = 0.5 |

- *Correct concept* **Pet** gets perfect score with all metrics
- Recall incorrectly gives perfect score to superclass *Animal*
- Precision incorrectly gives perfect score to subclasses *Dog* and *Cat*

*Figure B.1.* A hypothetical neuron that only activates on pets (dogs or cats). When comparing different evaluation metrics, we can see recall cannot distinguish between the correct concept (Pet) and a concept that is too generic (Animal), while precision favors concepts that are too specific (Dog, Cat). IoU can unambiguously determine the correct concept.

## B.4. Additional Related Works

**Evaluation of individual neuron explanations.**    Table B.1 shows an expanded version of our comparison table (Table 1 in the main text section 4) of existing evaluation methods. It can be seen in Table B.1 that prior work (Bau et al., 2017; Oikarinen & Weng, 2023; Oikarinen et al., 2023; Bai et al., 2025; Huang et al., 2023; Gurnee et al., 2023; Mu & Andreas, 2020; La Rosa et al., 2024; Koh et al., 2020; Zimmermann et al., 2023; Bykov et al., 2023; Kopf et al., 2024; Bills et al., 2023; Oikarinen & Weng, 2024; Bricken et al., 2023; Templeton et al., 2024; Oikarinen et al., 2023; Oikarinen & Weng, 2023; Shaham et al., 2024; Singh et al., 2023) only use 1-2 metrics for evaluation and typically did not discuss or justify why the metric should be used. Among all the prior work in Table B.1, we believe that the most similar work to ours is (Huang et al., 2023), which has focused on evaluating individual neuron explanations in language models. In particular, they discover a discrepancy between the evaluation metrics, i.e. neurons with very high Correlation(top-and-random) score can still have relatively low F1-scores. However, different from our work their scope is much more specific and they do not provide analysis comparing different evaluation metrics or justification on why they use F1-score specifically. Their findings are in line with ours, as we found that Correlation(top-and-random) fails the extra labels test, while F1-score passes our sanity checks.

Overall we find many evaluations in previous works to be lacking, either due to using poor metrics that fail our sanity checks, or using very small sample sizes. In addition, some popular methods like TCAV (Kim et al., 2018) completely lack evaluation of whether the concept directions they learn are good. To our best knowledge, no human-study has been conducted using metrics that pass our sanity checks, instead most existing human-studies only measure Recall or a similar metric, and running such a study would be valuable for better understanding of unit interpreability and/or explanation methods.

**Known Concept Evaluation:** Similar to our experiments in Section 5, many previous works such as (Oikarinen & Weng, 2023; Schwettmann et al., 2023; Moakhar et al., 2024; Shaham et al., 2024) have utilized neurons or units with known concepts to evaluate explanation quality. However, these papers typically focus on evaluating explanation methods or individual explanations, while our focus is on *meta-evaluation* to understand which metrics are best to use for evaluation, meaning our methods and motivations for the evaluation are noticeably different from these previous works.

**Interpretability Illusions:**    (Bolukbasi et al., 2021) discovered an interpretability illusion for BERT-models, which shows that neurons that look interpretable when only looking at their most highly activating inputs on one probing dataset $\mathcal{D}$, seem to be doing something completely different when analyzed on another dataset. This highlights the importance of good choice of probing dataset, or confirming explanations on multiple datasets. However, we believe there is another reason for these illusions, namely that the authors only look at highly activating inputs (equivalent to Recall in our framework), which is not a reliable metric. Overall, our goal is to reduce such interpretability illusions, i.e. the appearance of interpretability when something is really not interpretable, by encouraging more rigorous evaluation.

**Sanity Checks.**    The sanity checks proposed in this paper (Missing and Extra Labels test) are inspired by the sanity checks (Adebayo et al., 2018) proposed for saliency maps, which has had a large impact in guiding that field towards more faithful explanations. However the topic is very different, since (Adebayo et al., 2018) focus on local input-importance instead of global neuron explanations in our paper. Similarly, the specific tests proposed by (Adebayo et al., 2018) are also very different from ours.

**Concept extraction.**    Extracting interpretable concepts from a learned representation is a common challenge and relevant for finding individual units to evaluate in our framework. This can either be supervised as proposed by (Kim et al., 2018) where the concepts are specified by human and labels are provided. This approach is also used by linear probing based work such as (Gurnee et al., 2023). Later, a series of works(Ghorbani et al., 2019; Zhang et al., 2021; Fel et al., 2023) was proposed to automatically extract concepts from model activations, without human supervision, i.e. unsupervised. (Fel et al., 2024) claimed that concept extraction could be regarded as a dictionary learning problem. Recently Sparse Autoencoders (Cunningham et al., 2023; Templeton et al., 2024) have also gained popularity as an unsupervised concept extraction method. Note that most unsupervised concept extraction methods are discovering "units" defined in our work that are not directly understandable to humans, and require an explanation method to provide human-understandable explanations. Our work focuses on the evaluation of the explanations of those concepts.

| Metric $M$ | Study | Concept Source $c_t$ | Granularity | Probing Dataset $\mathcal{D}$ | Domain | Target |
|---|---|---|---|---|---|---|
| Recall | Human Eval (Zhou et al., 2015) | Crowdsourced | ∼ Whole Input | Specific eval dataset | Vision | Neurons |
| ∼Recall | Human Eval (Bau et al., 2017) | Crowdsourced | ∼Whole Input | Specific eval dataset | Vision | Neurons |
| ∼Recall | Human Eval (Oikarinen & Weng, 2023) (Bai et al., 2025; Srinivas et al., 2025) | Crowdsourced | Whole Input | Validation data | Vision | Neurons |
| ∼Recall | Human Eval (Oikarinen et al., 2023) | Crowdsourced | Whole Input | Validation Data | Vision | CBM neurons |
| Precision | Automated Eval (Srinivas et al., 2025) | Generative | Whole Input | Generated+ Validation data | Vision | Neurons |
| F1-score | Observation based (Huang et al., 2023) | Generative + model | Whole Input | Generated + Training data | Language | Neurons |
| F1-score | Sparse probes (Gurnee et al., 2023) | Labeled data | Per-token | Specific eval dataset | Language | Linear comb. of neurons |
| F1-score | CBM - Concept Error (Fig. 2) (Koh et al., 2020) | Labeled data | Whole Input | Validation Data | Vision | CBM neurons |
| IoU | Broden IoU (Bau et al., 2017) (Mu & Andreas, 2020; La Rosa et al., 2024) | Labeled data | Per-pixel | Specific eval dataset | Vision | Neurons |
| Accuracy | CBM - Concept Error (Koh et al., 2020) | Labeled data | Whole Input | Validation Data | Vision | CBM neurons |
| ∼AUC | Comparative Human Study (Zimmermann et al., 2023) | Crowdsourced | Whole Input | Training data | Vision | Neurons |
| Inverse AUC | INVERT (Bykov et al., 2023) | Labeled data | Whole Input | Validation data | Vision | Neurons |
| Inverse AUC | CoSy AUC (Kopf et al., 2024) | Generative | Whole Input | Generated+ Validation data | Vision | Neurons |
| Correlation(T&R) | Simulation - Correlation Score (Bills et al., 2023) | Model | Per-token | Training data | Language | Neurons |
| Correlation | Simulation - Correlation Score (Oikarinen & Weng, 2024) | Model | Whole Input | Validation data | Vision | Neurons |
| Correlation | CBM - Concept Error (Fig. 2) (Koh et al., 2020) | Labeled data | Whole Input | Validation Data | Vision | CBM neurons |
| Spearman Correlation(T&R) | SAE Auto Interp (Bricken et al., 2023; Templeton et al., 2024) | Model | Per-token | Training data | Language | SAE features |
| ∼WPMI | CLIP-Dissect - Similarity (Oikarinen & Weng, 2023) | Model | Whole Input | Validation data | Vision | Neurons |
| MAD | CoSy MAD (Kopf et al., 2024) | Generative | Whole Input | Generated+ Validation data | Vision | Neurons |
| ∼MAD | MAIA (Shaham et al., 2024) | Generative | Whole Input | Generated | Vision | Neurons |
| ∼MAD | Explanation Score (Singh et al., 2023) | Generative | Whole Input | Generated | Language | Transformer factors |

*Table B.1.* Extended Table (of Table 1 in the main text section 4) comparing related work. Note (Zhou et al., 2015) call their evaluation Precision, as they use the classification framing. This is equivalent to Recall in our framing as shown in Section A.2.

**Human-centered evaluation on concept-based models.** Human-centered evaluation of model explanation(Kim et al., 2024; Boyd-Graber et al., 2022) has drawn attention from the XAI community. Recently, (Kazmierczak et al., 2024) collected human evaluation of XAI explanations as a benchmark for explanation methods. These works provide important techniques and inspiration for evaluating explainability, but almost all existing work is focus on local feature importance explanations, which is very different from our work on global neuron-level explanations.

**Evaluation Metrics for Input-Improtance Explanations** The field of input-importance explanations has seen an evolution in the evaluation metrics used, with initial focus on finding the features that humans think are important. Later metrics such as deletion and insertion proposed by (Petsiuk et al., 2018) allow for more principled evaluation of the explanation fidelity, i.e. whether it actually matches what the model in vision models, and (Fel et al., 2021) extends the deletion metric to natural language settings. (Hooker et al., 2019) proposes the Remove And Retrain (ROAR) framework as an alternative method for evaluating the quality of input-importance explanations by evaluating whether a model retrained on data without the most important pixels can still solve the task. (Bhatt et al., 2020) propose additional checks, such as measuring whether an explanation method has high sensitivity, with the intuition that similar inputs should have similar explanations. While this line of work focuses on a different topic (local input importance explanations) instead of global neuron-level explanations, and mostly proposes new metrics without significant meta-evaluation, it highlights the importance of good evaluation mnetrics in explainable AI.

## C. Theoretical Missing/Extra Labels Test and Concept Imbalance

In this section we analyze the effects of missing/extra labels test on a simpler toy setting, where neuron $k$'s activations $a_k$ perfectly match the concept labels of concept $t_k$, i.e. $a_k = c_{t_k}$. In this setting, we assume binary neuron and concept activations, i.e. the neuron's activation is 1 if concept is present on the input, and 0 if its not. Consequently, we do not need to perform additional binarization of concept activations with top $\alpha$ like we did in previous sections.

In this simplified setting, our missing and extra labels tests correspond to being able to differentiate between three concepts as defined in section 4:

1. $c_{t_k}$: The perfect predictor for neuron $a_k$, with precision=1 and recall=1.

2. $c_{t_k}^-$: (Missing labels) This concept has Precision=1 since whenever a concept is present, the neuron is also active, and Recall of 0.5 since only half to inputs where the neuron activate now have the concept.

3. $c_{t_k}^+$: (Extra labels) Inverse from above, this concept has Precision=0.5 and Recall=1.

We then measure the average score difference $\Delta s$ across neurons:

- Missing labels test: $\Delta s(k) = \mathbb{E}_{c_{t_k}^-} [s_M(a_k, c_{t_k}^-) - s_M(a_k, c_{t_k})]$

- Extra labels test: $\Delta s(k) = \mathbb{E}_{c_{t_k}^+} [s_M(a_k, c_{t_k}^+) - s_M(a_k, c_{t_k})]$

A good metric should be able to reliably differentiate between these concepts. Interestingly we find that the ability of most metrics to differentiate greatly depends on whether the data is balanced or not.

Since the neuron activations perfectly align with concept $t$ and are binary, the only parameter that can effect the results of our missing and extra labels test is the activation frequency of concept $t$, i.e. what fraction of inputs $x \in \mathcal{D}$ contain concept $t$. Following the notation in section D, we denote this fraction as $\gamma$. Note technically it should be $\gamma + \eta$, but $\eta = 0$ in this case with perfect match between concept and neuron. We then test whether a metric passes the test on different values of $\gamma$, using simulated data on tables C.3 and C.4. Each number is the average result from 1000 evaluations with 500,000 datapoints each. In addition, in Section D, we derived a closed form solution to the binary metrics under missing or extra labels as a function of $\gamma$ and other parameters. This simplifies nicely when we consider an ideal neuron with $a_k = c_t$, and we can derive the expected result of missing/extra labels test as a simple function of $\gamma$ alone in Table D.2. These theoretical results perfectly agree with our simulated results.

### C.1. Results

In Tables C.1 and C.2 we report the Decrease Acc defined in Equation 11 on different activation frequencies $\gamma$ in our simulated neuron. We say a metric as passing the Theoretical test if it has Decrease Acc > 90% on every frequency. In tables C.3 and C.4 (visualized in Fig. 3) we report the average $\Delta s$ on the same set of different activation sparsities. We have highlighted metrics with $\Delta s > -0.01$ in red to highlight metrics approaching 0 difference.

We can see most metrics (expect from recall and precision) perform well on balanced data ($\gamma = 0.499$ and $\gamma = 0.1$). However, their performance often starts to drop with score difference $\Delta s$ approaching 0 as the data becomes more and more unbalanced. We can see that practically all the metrics that failed our experimental missing/extra labels tests cannot differentiate between perfect and modified concept specifically on imbalanced data, highlighting that likely the root cause of the failure on these test is that the metrics performs poorly on imbalanced data. This is also aligned with conventional knowledge that metrics such as accuracy and AUC are a poor choice to rely on when your data is heavily imbalanced. On the other hand, metrics that passed the tests are insensitive to activation frequency $\gamma$ and converge to a nonzero constant as $\gamma$ decreases.

The results in terms of passing are almost identical to our experimental results in section 4. The only differences were WPMI which passes the theoretical extra labels test but fails the experimental one. We believe this has to do with hyperparameter($\alpha, \lambda$) choices and that WPMI can in principle pass the test but with poor hyperparameters(small $\lambda$) it will not, leading us to overall recommend against using it in practice as hyperparameter choice is challenging in the real world. An interesting case is Inverse AUPRC. This metric is designed for imbalanced data and works well in that domain,

but actually performs worse when the data is balanced. In particular Inverse AUPRC fails the test when data is perfectly balanced. This indicates caution should be used if relying on it, in case you have some neurons with extremely common concepts.

We argue that being able to pass these tests regardless of activation frequency is important for any evaluation metric to be used, as we typically do not know what frequency each neuron will have in advance, in many cases the interesting neurons/concept might activate very sparsely, for example in Sparse Autoencoders.

| Theoretical Missing Labels Test, Decrease Acc | | | | | | |
|---|---|---|---|---|---|---|
| **Activation Frequency** $\gamma$**:** | 0.499 | 0.1 | 0.01 | 0.001 | 0.0001 | Pass |
| **Recall** | 100.00% | 100.00% | 100.00% | 100.00% | 100.00% | ✓ |
| **Precision** | 0.00% | 0.00% | 0.00% | 0.00% | 0.00% | ✗ |
| **F1-score** | 100.00% | 100.00% | 100.00% | 100.00% | 100.00% | ✓ |
| **IoU** | 100.00% | 100.00% | 100.00% | 100.00% | 100.00% | ✓ |
| **Accuracy** | 100.00% | 100.00% | 100.00% | 0.00% | 0.00% | ✗ |
| **Balanced Acc.** | 100.00% | 100.00% | 100.00% | 100.00% | 100.00% | ✓ |
| **Inverse Balanced Acc.** | 100.00% | 100.00% | 100.00% | 0.00% | 0.00% | ✗ |
| **AUC** | 100.00% | 100.00% | 100.00% | 100.00% | 100.00% | ✓ |
| **Inverse AUC** | 100.00% | 100.00% | 100.00% | 0.00% | 0.00% | ✗ |
| **Correlation** | 100.00% | 100.00% | 100.00% | 100.00% | 100.00% | ✓ |
| **Correlation(T&R)** | 100.00% | 100.00% | 100.00% | 100.00% | 100.00% | ✓ |
| **Spearman Correlation** | 100.00% | 100.00% | 100.00% | 22.70% | 13.30% | ✗ |
| **Spearman Correlation(T&R)** | 100.00% | 100.00% | 100.00% | 100.00% | 100.00% | ✓ |
| **Cosine** | 100.00% | 100.00% | 100.00% | 100.00% | 100.00% | ✓ |
| **WPMI** | 100.00% | 100.00% | 100.00% | 100.00% | 100.00% | ✓ |
| **MAD** | 100.00% | 100.00% | 100.00% | 0.00% | 0.00% | ✗ |
| **AUPRC** | 100.00% | 100.00% | 100.00% | 100.00% | 100.00% | ✓ |
| **Inverse AUPRC** | 100.00% | 100.00% | 100.00% | 100.00% | 100.00% | ✓ |

*Table C.1.* Average Decrease Accuracy on Theoretical Missing labels Test. A metric fails if it has low decrease accuracy on any concept frequency $\gamma$.

| Theoretical Extra Labels Test, Decrease Acc | | | | | | |
|---|---|---|---|---|---|---|
| **Activation Frequency $\gamma$:** | 0.499 | 0.1 | 0.01 | 0.001 | 0.0001 | Pass |
| **Recall** | 0.00% | 0.00% | 0.00% | 0.00% | 0.00% | ✗ |
| **Precision** | 100.00% | 100.00% | 100.00% | 100.00% | 100.00% | ✓ |
| **F1-score** | 100.00% | 100.00% | 100.00% | 100.00% | 100.00% | ✓ |
| **IoU** | 100.00% | 100.00% | 100.00% | 100.00% | 100.00% | ✓ |
| **Accuracy** | 100.00% | 100.00% | 100.00% | 48.40% | 0.00% | ✗ |
| **Balanced Acc.** | 100.00% | 100.00% | 100.00% | 0.00% | 0.00% | ✗ |
| **Inverse Balanced Acc.** | 100.00% | 100.00% | 100.00% | 100.00% | 100.00% | ✓ |
| **AUC** | 100.00% | 100.00% | 100.00% | 0.00% | 0.00% | ✗ |
| **Inverse AUC** | 100.00% | 100.00% | 100.00% | 100.00% | 100.00% | ✓ |
| **Correlation** | 100.00% | 100.00% | 100.00% | 100.00% | 100.00% | ✓ |
| **Correlation(T&R)** | 100.00% | 92.80% | 22.60% | 2.70% | 0.10% | ✗ |
| **Spearman Correlation** | 100.00% | 100.00% | 5.60% | 3.20% | 11.60% | ✗ |
| **Spearman Correlation(T&R)** | 0.20% | 8.60% | 20.60% | 63.70% | 5.30% | ✗ |
| **Cosine** | 100.00% | 100.00% | 100.00% | 100.00% | 100.00% | ✓ |
| **WPMI** | 100.00% | 100.00% | 100.00% | 100.00% | 100.00% | ✓ |
| **MAD** | 100.00% | 100.00% | 100.00% | 100.00% | 100.00% | ✓ |
| **AUPRC** | 100.00% | 100.00% | 100.00% | 100.00% | 100.00% | ✓ |
| **Inverse AUPRC** | 47.70% | 100.00% | 100.00% | 100.00% | 100.00% | ✗ |

*Table C.2.* Average Decrease Accuracy on Theoretical Extra labels Test. A metric fails if it has low decrease accuracy on any concept frequency $\gamma$.

| Theoretical Missing Labels Test, Average $\Delta s$ | | | | | | |
|---|---|---|---|---|---|---|
| **Activation Frequency $\gamma$:** | 0.499 | 0.1 | 0.01 | 0.001 | 0.0001 | $\lim_{\gamma \to 0}$ |
| **Recall** | -0.5000 | -0.4999 | -0.5002 | -0.5007 | -0.5025 | -0.5000 |
| **Precision** | 0.0000 | 0.0000 | 0.0000 | 0.0000 | 0.0000 | 0.0000 |
| **F1-score** | -0.3334 | -0.3333 | -0.3335 | -0.3341 | -0.3352 | -0.3333 |
| **IoU** | -0.5000 | -0.5002 | -0.4998 | -0.5005 | -0.5032 | -0.5000 |
| **Accuracy** | -0.2495 | -0.0500 | -0.0050 | -0.0005 | 0.0000 | 0.0000 |
| **Balanced Acc.** | -0.2500 | -0.2500 | -0.2501 | -0.2500 | -0.2500 | -0.2500 |
| **Inverse Balanced Acc.** | -0.1662 | -0.0263 | -0.0025 | -0.0002 | 0.0000 | 0.0000 |
| **AUC** | -0.2500 | -0.2500 | -0.2500 | -0.2493 | -0.2508 | - |
| **Inverse AUC** | -0.1662 | -0.0263 | -0.0025 | -0.0003 | 0.0000 | - |
| **Correlation** | -0.2111 | -0.1559 | -0.1474 | -0.1466 | -0.1479 | - |
| **Correlation(T&R)** | -0.4965 | -0.4894 | -0.4649 | -0.3928 | -0.2449 | - |
| **Spearman Correlation** | -0.2000 | -0.0686 | -0.0079 | -0.0006 | -0.0018 | - |
| **Spearman Correlation(T&R)** | -0.1005 | -0.1009 | -0.1066 | -0.0902 | -0.2006 | - |
| **Cosine** | -0.1464 | -0.1465 | -0.1465 | -0.1464 | -0.1474 | - |
| **WPMI** | -0.3590 | -0.3591 | -0.3589 | -0.3587 | -0.3602 | - |
| **MAD** | -0.2210 | -0.0350 | -0.0033 | -0.0003 | 0.0000 | - |
| **AUPRC** | -0.2505 | -0.4499 | -0.4953 | -0.5003 | -0.4964 | - |
| **Inverse AUPRC** | -0.5000 | -0.4999 | -0.5002 | -0.4988 | -0.4996 | - |

*Table C.3.* Simulation results missing labels test on idealized neuron with perfect correspondence to a concept activation. We can see most metrics pass when the data is relatively balanced, but several start to struggle on inbalanced data (low $\gamma$). In contrast, passing metrics maintain a clearly nonzero $\Delta s$ regardless of activation frequency. The $\lim_{\gamma \to 0}$ column is analytical solution from D.2.

| Theoretical Extra Labels Test, Average $\Delta s$ | | | | | | |
|---|---|---|---|---|---|---|
| **Activation Frequency $\gamma$:** | 0.499 | 0.1 | 0.01 | 0.001 | 0.0001 | $\lim_{\gamma \to 0}$ |
| **Recall** | 0.0000 | 0.0000 | 0.0000 | 0.0000 | 0.0000 | 0.0000 |
| **Precision** | -0.5000 | -0.5001 | -0.4999 | -0.4993 | -0.4963 | -0.5000 |
| **F1-score** | -0.3333 | -0.3334 | -0.3334 | -0.3336 | -0.3333 | -0.3333 |
| **IoU** | -0.5000 | -0.5000 | -0.5001 | -0.4997 | -0.4970 | -0.5000 |
| **Accuracy** | -0.4990 | -0.1000 | -0.0100 | -0.0010 | -0.0001 | 0.0000 |
| **Balanced Acc.** | -0.4980 | -0.0556 | -0.0051 | -0.0005 | 0.0000 | 0.0000 |
| **Inverse Balanced Acc.** | -0.2500 | -0.2500 | -0.2500 | -0.2500 | -0.2483 | -0.2500 |
| **AUC** | -0.7562 | -0.0844 | -0.0077 | -0.0008 | -0.0001 | - |
| **Inverse AUC** | -0.2500 | -0.2500 | -0.2501 | -0.2500 | -0.2491 | - |
| **Correlation** | -0.4777 | -0.1667 | -0.1483 | -0.1465 | -0.1461 | - |
| **Correlation(T&R)** | -0.4929 | -0.0293 | -0.0027 | -0.0002 | 0.0000 | - |
| **Spearman Correlation** | -0.5672 | -0.0238 | -0.0007 | -0.0002 | -0.0018 | - |
| **Spearman Correlation(T&R)** | 0.0002 | 0.0000 | -0.0036 | 0.0307 | -0.0064 | - |
| **Cosine** | -0.1464 | -0.1464 | -0.1463 | -0.1464 | 0.1456 | - |
| **WPMI** | -0.0292 | -0.0292 | -0.0292 | -0.0293 | -0.0292 | - |
| **MAD** | -0.3324 | -0.3324 | -0.3324 | -0.3321 | -0.3298 | - |
| **AUPRC** | -0.5000 | -0.5000 | -0.4999 | -0.4998 | -0.4974 | - |
| **Inverse AUPRC** | -0.0010 | -0.4000 | -0.4899 | -0.4984 | -0.4957 | - |

*Table C.4.* Simulation results extra labels test on idealized neuron with perfect correspondence to a concept activation. We can see most metrics pass when the data is relatively balanced, but several start to struggle on inbalanced data (low $\gamma$). In contrast, passing metrics maintain a clearly nonzero $\Delta s$ regardless of activation frequency. The $\lim_{\gamma \to 0}$ column is analytical solution from D.2.

# D. Analytical solution for Missing/Extra Labels Test

In this section, we provide a theoratical analysis for missing label and extra label test for binary classification metrics. For simplicity of symbols, in this section we analyze the population statistics. Suppose we have following confusion matrix:

|       | c=1      | c=0 |
|-------|----------|-----|
| a=1   | $\gamma$ | $b$ |
| a=0   | $\eta$   | $d$ |

Here, we use $a$ to denote neuron activation and $c$ to denote concept activation. In missing labels test, consider a general case where we randomly flip $c = 1$ into $c = 0$ with probability $p$. Thus, the resulting confusion matrix is:

|       | c=1             | c=0             |
|-------|-----------------|-----------------|
| a=1   | $(1-p)\gamma$   | $b + p\gamma$   |
| a=0   | $(1-p)\eta$     | $d + p\eta$     |

In extra labels test, similarly, we turn $c = 0$ into $c = 1$ with probability $q = \frac{p(\gamma+\eta)}{b+d}$, the resulting confusion matrix is

|       | c=1             | c=0          |
|-------|-----------------|--------------|
| a=1   | $\gamma + qb$   | $(1-q)b$     |
| a=0   | $\eta + qd$     | $(1-q)d$     |

With these, we could plug in corresponding TP/FP/TN/FN into metrics to calculate metric value in these two tests.

1. **Recall:**

$$s_M(a_k, c_t) = \frac{B(a_k) \cdot B(c_t)}{||B(a_k)||_1} = \frac{\gamma}{b + \gamma}. \tag{D.1}$$

   In extra label test:

$$s_M(a_k, c_t^+) = \frac{\gamma + qb}{b + \gamma} \geq s_M(a_k, c_t). \tag{D.2}$$

   In missing label test:

$$s_M(a_k, c_t^-) = \frac{\gamma - p\gamma}{b + \gamma} \leq s_M(a_k, c_t). \tag{D.3}$$

   From the derivation above, we could see that increasing labels only raises recall metric while reducing labels always leads to a drop in recall as we found in our experiments.

2. **Precision:**

$$s_M(a_k, c_t) = \frac{B(a_k) \cdot B(c_t)}{||B(c_t)||_1} = \frac{\gamma}{\gamma + \eta} \tag{D.4}$$

   In extra label test:

$$s_M(a_k, c_t^+) = \frac{\gamma + qb}{\gamma + qb + \eta + qd}. \tag{D.5}$$

   Precision will increase if $\frac{b}{b+d} > \frac{\gamma}{\gamma+\eta}$.

   In missing label test:

$$s_M(a_k, c_t^-) = \frac{(1-p)\gamma}{(1-p)\gamma + (1-p)\eta} = s_M(a_k, c_t). \tag{D.6}$$

   Thus, the precision does not change in the missing labels test.

3. **F1-score:**

$$s_M(a_k, c_t) = \frac{2\gamma}{2\gamma + \eta + b} \tag{D.7}$$

In extra label test:

$$s_M(a_k, c_t^+) = \frac{2\gamma + 2qb}{2\gamma + qb + \eta + qd + b}. \tag{D.8}$$

F1-score increases if $\frac{2b}{b+d} > \frac{2\gamma}{2\gamma + \eta + b}$.

In missing label test:

$$s_M(a_k, c_t^-) = \frac{2(1-p)\gamma}{2(1-p)\gamma + (1-p)\eta + b + p\gamma} = \frac{2\gamma - 2p\gamma}{2\gamma + (1-p)\eta + b - p\gamma}. \tag{D.9}$$

F1-score decreases in missing label test.

4. **IoU:**

$$s_M(a_k, c_t) = \frac{B(a_k) \cdot B(c_t)}{||B(a_k)||_1 + ||B(c_t)||_1 - B(a_k) \cdot B(c_t)} = \frac{\gamma}{\gamma + \eta + b}. \tag{D.10}$$

In extra label test:

$$s_M(a_k, c_t^+) = \frac{\gamma + qb}{\gamma + \eta + qd + b}. \tag{D.11}$$

IoU increases if $\frac{b}{d} > \frac{\gamma}{\gamma + \eta + b}$.

In missing label test:

$$s_M(a_k, c_t^-) = \frac{(1-p)\gamma}{(1-p)\gamma + (1-p)\eta + b + p\gamma} = \frac{\gamma - p\gamma}{\gamma + (1-p)\eta + b}. \tag{D.12}$$

Thus, IoU decreases in missing label test.

5. **Accuracy:**

$$s_M(a_k, c_t) = \frac{B(a_k) \cdot B(c_t) + (\mathbf{1} - B(a_k)) \cdot (\mathbf{1} - B(c_t))}{n} = \gamma + d. \tag{D.13}$$

In extra label test:

$$s_M(a_k, c_t^+) = \gamma + qb + d - qd. \tag{D.14}$$

accuracy increases if $b > d$.

In missing label test:

$$s_M(a_k, c_t^-) = \gamma - p\gamma + d + p\eta. \tag{D.15}$$

Accuracy increases if $\eta > \gamma$.

6. **Balanced Accuracy:**

$$s_M(a_k, c_t) = \frac{B(a_k) \cdot B(c_t)}{2||B(a_k)||_1} + \frac{(\mathbf{1} - B(a_k)) \cdot (\mathbf{1} - B(c_t))}{2||(1 - B(a_k))||_1} = \frac{\gamma}{2\gamma + 2b} + \frac{d}{2\eta + 2d}. \tag{D.16}$$

In extra label test:

$$s_M(a_k, c_t^+) = \frac{\gamma + qb}{2\gamma + 2b} + \frac{d - qd}{2\eta + 2d}. \tag{D.17}$$

balanced accuracy increases if $\frac{b}{2\gamma + 2b} > \frac{d}{2\eta + 2d}$.

In missing label test:

$$s_M(a_k, c_t^-) = \frac{\gamma - p\gamma}{2\gamma + 2b} + \frac{d + p\eta}{2\eta + 2d}. \tag{D.18}$$

balanced accuracy increases if $\frac{\gamma}{2\gamma + 2b} < \frac{\eta}{2\eta + 2d}$.

7. **Inverse Balanced Accuracy:**

$$s_M(a_k, c_t) = \frac{B(a_k) \cdot B(c_t)}{2||B(c_t)||_1} + \frac{(\mathbf{1} - B(a_k)) \cdot (\mathbf{1} - B(c_t))}{2||(\mathbf{1} - B(c_t))||_1} = \frac{\gamma}{2\gamma + 2\eta} + \frac{d}{2b + 2d}. \tag{D.19}$$

In extra label test:

$$s_M(a_k, c_t^+) = \frac{\gamma + qb}{2\gamma + 2\eta + 2qb + 2qd} + \frac{d - qd}{2b + 2d - 2qd - 2qb}. \tag{D.20}$$

In missing label test:

$$s_M(a_k, c_t^-) = \frac{\gamma - p\gamma}{2\gamma + 2\eta - 2p\gamma - 2p\eta} + \frac{d + p\eta}{2b + 2d + 2p\gamma + 2p\eta}. \tag{D.21}$$

| Metric | Missing label: $s_M(a_k, c_t^-)$ | Extra label: $s_M(a_k, c_t^+)$ |
|---|---|---|
| Recall | $1 - p$ | $1$ |
| Precision | $1$ | $\frac{1}{1+p}$ |
| F1-score | $\frac{2-2p}{2-p}$ | $\frac{2}{2+p}$ |
| IoU | $1 - p$ | $\frac{1}{1+p}$ |
| Accuracy | $1 - p\gamma$ | $1 - \gamma p$ |
| Balanced Accuracy | $1 - \frac{p}{2}$ | $1 - \frac{p\gamma}{2(1-\gamma)}$ |
| Inverse Balanced Accuracy | $\frac{(1-\gamma)}{2(1-\gamma)+2p\gamma} + \frac{1}{2}$ | $\frac{2+p}{2(1+p)}$ |

*Table D.1.* The evaluation scores for different metrics under missing and extra label tests on an *ideal* neuron whose activations perfectly match the presence of our concept.

## D.1. Special case - Theoretical Test

In this section, we consider a special case where the neuron activations perfectly match the concept, i.e. $c \equiv a$. In this case, we have $\eta = b = 0$, $d = 1 - \gamma$. Plugging in those variables, we get the equations for metric scores under sanity test settings in Table D.1. Further simplifying this by plugging in the $p$ values we typically use we get the simple form for score diff in Table D.2.

| Metric | Missing labels test($p = 0.5$): $s_M(a_k, c_t^-) - s_M(a_k, c_t)$ | Extra labels test($p = 1$): $s_M(a_k, c_t^+) - s_M(a_k, c_t)$ |
|---|---|---|
| Recall | $-0.5$ | $0$ |
| Precision | $0$ | $-0.5$ |
| F1-score | $-\frac{1}{3}$ | $-\frac{1}{3}$ |
| IoU | $-0.5$ | $-0.5$ |
| Accuracy | $-\frac{\gamma}{2}$ | $-\gamma$ |
| Balanced Accuracy | $-0.25$ | $-\frac{\gamma}{2(1-\gamma)}$ |
| Inverse Balanced Accuracy | $-\frac{\gamma}{2(2-\gamma)}$ | $-0.25$ |

*Table D.2.* Further simplifying from Table D.1 by plugging in $p$ values we typically use in our tests, and $s_M(a_k, c_t) = 1$, we can calculate the theoretical score diff after running our tests for different binary metrics on ideal neurons.

# E. Ablations and Extra checks

### E.1. Increase/decrease Fraction in Missing/Extra labels test

In our standard missing labels we reduce the fraction of positive labels by half, i.e.

$$r_- = \frac{||c_t^-||_1}{||c_t||} = 0.5$$

$$r_+ = \frac{||c_t^+||_1}{||c_t||_1} = 2$$

In this section we run an ablation on the importance of these specific values by running the test on two other combinations of values: $r_- = 0.75$ and $r_+ = 1.33$ (smaller change) and $r_- = 0.33$ and $r_+ = 3$ (larger change). We report the results on final layer neurons (including superclasses) of ViT-B-16 (trained on ImageNet) in Table E.1. We used ground turth $c_t$. Overall we can see that these parameters do not impact our qualitative observations, i.e. which metric passes vs which doesn't. While larger changes to $r$ typically increases Decrease Acc, it doesn't affect which metrics approach zero score diff vs which do not, which is what we're mostly interested in. Overall this shows our sanity test is not sensitive to the specific parameter choice but instead reflects overall trends of the metric.

| | Missing Labels Test | | | Extra Labels Test | | |
|---|---|---|---|---|---|---|
| | $r_- = 0.33$ | $r_- = 0.5$ | $r_- = 0.75$ | $r_+ = 1.33$ | $r_+ = 2.0$ | $r_+ = 3.0$ |
| **Recall** | 99.55% | 95.94% | 75.34% | 0.00% | 0.00% | 0.00% |
| **Precision** | 48.20% | 46.09% | 49.62% | 99.32% | 99.62% | 99.62% |
| **F1-score** | 97.07% | 95.64% | 92.78% | 99.62% | 99.70% | 99.70% |
| **IoU** | 96.84% | 95.34% | 92.41% | 99.32% | 99.62% | 99.62% |
| **Accuracy** | 0.00% | 0.00% | 0.00% | 16.24% | 63.08% | 99.92% |
| **Balanced Acc.** | 99.47% | 95.86% | 75.19% | 9.17% | 23.98% | 63.61% |
| **Inverse Balanced Acc.** | 48.05% | 45.56% | 48.35% | 99.17% | 99.62% | 99.92% |
| **AUC** | 98.65% | 95.34% | 76.32% | 16.99% | 30.23% | 59.32% |
| **Inverse AUC** | 24.51% | 21.35% | 16.47% | 99.92% | 99.92% | 99.92% |
| **Correlation** | 99.92% | 99.92% | 99.77% | 99.92% | 99.92% | 99.92% |
| **Correlation(T&R)** | 99.40% | 98.12% | 90.68% | 43.76% | 44.96% | 48.12% |
| **Spearman Correlation** | 57.29% | 53.91% | 46.92% | 37.07% | 38.05% | 37.67% |
| **Spearman Correlation(T&R)** | 97.14% | 93.61% | 82.63% | 49.92% | 50.75% | 52.26% |
| **Cosine** | 100.00% | 100.00% | 99.85% | 99.85% | 99.85% | 99.85% |
| **WPMI** | 99.55% | 95.94% | 75.34% | 99.77% | 99.70% | 99.77% |
| **MAD** | 50.83% | 49.47% | 49.02% | 99.70% | 99.70% | 99.70% |
| **AUPRC** | 99.47% | 99.25% | 98.27% | 99.32% | 99.62% | 99.62% |
| **Inverse AUPRC** | 100.00% | 100.00% | 99.85% | 99.70% | 99.62% | 99.32% |

*Table E.1.* Ablation study on sanity checks varying the magnitude of change in labels. We can see overall our results are consistent and not heavily dependent on the choice of $r$

## E.2. Failure case of cosine: Adding a constant activation

In this section we discuss a failure case of cosine similarity as an evaluation metric. The main idea is that cosine similarity outputs are not independent of the mean of the neuron's activation, while all other evaluation metrics are. This causes it to associate neurons with large average activation values with very generic concepts that are almost always active. As an example, we added a constant (1) to all activations of concept neurons in a Concept Bottleneck Model (Koh et al., 2020) trained on CUB200, and ran our meta-AUPRC test(Sec 5). All other evaluation metrics are invariant to this change, but the AUPRC of Cosine drops from 99.07% to 0.93% with this small change, as shown in Table E.2. We believe this is a flaw that points against using cosine similarity, as the average activation of a hidden layer neuron is not functionally important, and could be absorbed into biases of the next layer. Instead we recommend using Pearson correlation which is identical to cosine similarity after normalizing to mean 0, as we show in Section A.4.

| CUB200 - CBM concept neurons | | | |
|---|---|---|---|
| | Original $a_k$ | $a'_k = a_k + 1$ | |
| **Metric** | **AUPRC** | **AUPRC** | **$\Delta$ AUPRC** |
| **Recall** | 0.3958 | 0.3958 | 0.0000 |
| **Precision** | 0.6803 | 0.6803 | 0.0000 |
| **F1-score/IoU** | 0.6386 | 0.6386 | 0.0000 |
| **Accuracy** | 0.5853 | 0.5853 | 0.0000 |
| **Balanced Acc.** | 0.7165 | 0.7165 | 0.0000 |
| **Inverse Balanced Acc.** | 0.6628 | 0.6628 | 0.0000 |
| **AUC** | 0.7045 | 0.7045 | 0.0000 |
| **Inverse AUC** | 0.8956 | **0.8956** | 0.0000 |
| **Correlation** | 0.8883 | 0.8883 | 0.0000 |
| **Correlation(T&R)** | 0.7170 | 0.7170 | 0.0000 |
| **Spearman Correlation** | 0.4233 | 0.4233 | 0.0000 |
| **Spearman Correlation(T&R)** | 0.4243 | 0.4243 | 0.0000 |
| **Cosine** | **0.8985** | 0.0443 | -0.8542 |
| **WPMI** | 0.7323 | 0.7323 | 0.0000 |
| **MAD** | 0.7760 | 0.7760 | 0.0000 |
| **AUPRC** | 0.6712 | 0.6712 | 0.0000 |
| **Inverse AUPRC** | 0.3975 | 0.3975 | 0.0000 |

*Table E.2.* Adding a constant to the activation values of all neurons causes the cosine similarity to perform very poorly, while other metrics are unchanged.

### E.3. Random vs Semantic subsets

Our missing/extra labels tests are a mathematical model of semantic sub/superclasses that can be run without knowledge of the semantics or relationships between concepts. To test this mathematical model, we conducted a version of extra labels test on the ImageNet dataset where instead of randomly sampling $c^+$, we used the smallest superclass of the concept according to WordNet hierachy as $c^+$, and a version of missing labels test where we used the largest subclass of the concept as $c^-$. As shown in new Table G.5, the results are essentially identical to using random sub/supersets, showcasing our random sub/superset is a good model of real semantic relationships for our purposes.

| | ViT-B-16 superclass neurons | | ViT-B-16 single class neurons | |
| | Missing Labels Test | | Extra Labels Test | |
| **Metric** | **Random Subsets** | **Semantic Subsets** | **Random Supersets** | **Semantic Supersets** |
|---|---|---|---|---|
| **Recall** | 96.32% | 92.89% | 0.00% | 0.00% |
| **Precision** | 44.21% | 43.68% | 100.00% | 99.79% |
| **F1-score** | 97.37% | 91.84% | 100.00% | 99.79% |
| **IoU** | 96.84% | 91.58% | 100.00% | 99.79% |
| **Accuracy** | 44.74% | 31.32% | 46.11% | 71.37% |
| **Balanced Acc.** | 99.47% | 97.37% | 0.00% | 64.53% |
| **Inverse Balanced Acc.** | 28.95% | 37.37% | 100.00% | 99.79% |
| **AUC** | 99.21% | 97.11% | 17.47% | 52.11% |
| **Inverse AUC** | 51.58% | 54.47% | 100.00% | 96.00% |
| **Correlation** | 100.00% | 100.00% | 100.00% | 99.58% |
| **Correlation(T&R)** | 100.00% | 99.21% | 47.47% | 70.11% |
| **Spearman Correlation** | 70.53% | 75.26% | 37.37% | 20.63% |
| **Spearman Correlation(T&R)** | 98.42% | 97.63% | 51.05% | 35.16% |
| **Cosine** | 100.00% | 100.00% | 100.00% | 99.58% |
| **WPMI** | 100.00% | 100.00% | 0.00% | 0.00% |
| **MAD** | 58.42% | 51.58% | 100.00% | 100.00% |
| **AUPRC** | 99.21% | 96.84% | 100.00% | 99.37% |
| **Inverse AUPRC** | 100.00% | 100.00% | 100.00% | 99.37% |

*Table E.3.* Comparison of $c_t^-$ derived as a random subclass of $c_t$ vs a real semantic subclass of $c_t$ derived from the WordNet hierarchy. The missing labels test was evaluated on all 400 superclass neurons created in the final layer, while the extra labels test was conducted on all 1000 individual class neurons. We can see that from the point of view of passing our tests the random subclass behaves almost exactly as a real semantic subclass.

## E.4. Epsilon ablation

In this section we study the effect of the choice of $\epsilon$ in Equation 10 on our experimental missing/extra labels test. Overall we can see in Table E.4 that while changing $\epsilon$ by changes results for some metrics, for the most part it does not affect which metrics pass or do not pass the test despite 2 orders of magnitude change in $\epsilon$, showing our tests are not very sensitive to this choice.

Prehaps more importantly, theoretically we do not think the $\epsilon$ choice is important. An alternative and perhaps more fundamental definition for our theoretical tests could be as follows: A metric passes the test if

$$\exists \epsilon > 0 \text{ s.t. } \forall \gamma > 0, \Delta s < -\epsilon \tag{E.1}$$

where $\Delta s$ is calculated with the theoretical missing labels test defined in Appendix C and $\gamma$ is the concept/neuron activation frequency. This definition is purely a mathematical property of the metric without any hyperparameters or ties to any setting, and as far as we know matches our theoretical test results of Table C.1 and Table C.2 that use a fixed $\epsilon = 0.001$.

The issue with this alternative definition is that, we can only prove a metric passes this version of the test if we have an analytical solution, which we have for binary classification metrics in Table D.2. This makes this definition hard to use in practice. We can see for example accuracy fails this test as $\lim_{\gamma \to 0} \Delta s = 0$, so no matter what $\epsilon$ we choose, $\exists \gamma$ s.t. $\Delta s > -\epsilon$. On the other hand, for F1-score will pass the theoretical tests for any $\epsilon < 1/3$ regardless of $\gamma$. From this we can see that if we decrease $\epsilon$ in the current theoretical test, more metrics would pass, but the same metrics would fail again if we expand Table C.1/C.2 to the right by including smaller values, while current passing metrics would not fail regardless of how many additional $\gamma$ values we test.

| | Setting 1: ViT-B-16 Final Layer neurons | | | | | |
| --- | --- | --- | --- | --- | --- | --- |
| | **Missing Labels Test - Decrease Acc** | | | **Extra Labels Test - Decrease Acc** | | |
| | $\epsilon = 0.0001$ | $\epsilon = 0.001$ | $\epsilon = 0.01$ | $\epsilon = 0.0001$ | $\epsilon = 0.001$ | $\epsilon = 0.01$ |
| **Recall** | 97.07% | 97.07% | 97.07% | 0.00% | 0.00% | 0.00% |
| **Precision** | 48.65% | 47.14% | 37.74% | 99.70% | 99.20% | 97.89% |
| **F1-score** | 96.48% | 95.79% | 93.38% | 99.77% | 99.70% | 98.72% |
| **IoU** | 96.69% | 95.41% | 92.48% | 99.70% | 99.62% | 97.89% |
| **Accuracy** | 66.09% | 0.00% | 0.00% | 99.92% | 61.50% | 6.32% |
| **Balanced Acc.** | 97.07% | 97.07% | 97.07% | 99.77% | 23.08% | 3.76% |
| **Inverse Balanced Acc.** | 51.95% | 46.54% | 31.35% | 99.92% | 99.62% | 96.77% |
| **AUC** | 96.17% | 95.86% | 95.19% | 84.81% | 29.25% | 11.58% |
| **Inverse AUC** | 78.35% | 21.43% | 3.68% | 99.92% | 99.92% | 99.92% |
| **Correlation** | 100.00% | 100.00% | 99.92% | 99.92% | 99.92% | 99.92% |
| **Correlation(T&R)** | 97.52% | 97.29% | 96.47% | 48.42% | 46.02% | 37.07% |
| **Spearman Correlation** | 63.68% | 53.91% | 5.94% | 48.87% | 37.07% | 1.28% |
| **Spearman Correlation(T&R)** | 93.91% | 93.76% | 92.56% | 51.73% | 50.90% | 44.36% |
| **Cosine** | 100.00% | 100.00% | 100.00% | 99.92% | 99.85% | 99.70% |
| **WPMI** | 97.07% | 97.07% | 97.07% | 99.77% | 99.70% | 99.55% |
| **MAD** | 52.78% | 50.75% | 26.32% | 99.70% | 99.70% | 99.70% |
| **AUPRC** | 99.77% | 99.40% | 97.59% | 99.70% | 99.62% | 97.89% |
| **Inverse AUPRC** | 100.00% | 100.00% | 100.00% | 99.62% | 99.62% | 99.62% |

*Table E.4.* An Ablation study on the effect of $\epsilon$ choice on our results. While the exact percentages vary, we can see that changing $\epsilon$ by orders of magnitude mostly does not change which metrics pass the tests, highlighting robustness to this hyperparameter choice.

## E.5. Number of samples for $c_t^{\pm}$

Most of our results only use one sample of $c_t^{\pm}$ to reduce computational cost, even though our equations are defined in terms of its expectation. In Table E.5 we test whether using more samples of $c_t^{\pm}$ affects our results. As we can see, the number of samples has very little effect on our results likely because they are already averages over many neurons and large datasets.

| | Setting 1: ViT-B-16 Final Layer neurons | | | | | | | |
| --- | --- | --- | --- | --- | --- | --- | --- | --- |
| | Missing Labels Test | | | | Extra Labels Test | | | |
| $c_t^{\pm}$ samples per neuron: | 1 | 3 | 10 | 100 | 1 | 3 | 10 | 100 |
| Recall | 97.07% | 100.00% | 100.00% | 100.00% | 0.00% | 0.00% | 0.00% | 0.00% |
| Precision | 47.14% | 45.04% | 44.29% | 38.80% | 99.20% | 99.62% | 99.62% | 99.62% |
| F1-score | 95.79% | 96.77% | 97.29% | 97.74% | 99.70% | 99.70% | 99.70% | 99.70% |
| IoU | 95.41% | 96.02% | 96.62% | 97.22% | 99.62% | 99.62% | 99.62% | 99.62% |
| Accuracy | 0.00% | 0.00% | 0.00% | 0.00% | 61.50% | 61.88% | 64.66% | 66.54% |
| Balanced Acc. | 97.07% | 99.92% | 99.77% | 99.85% | 23.08% | 22.48% | 23.31% | 22.18% |
| Inverse Balanced Acc. | 46.54% | 43.83% | 42.41% | 34.44% | 99.62% | 99.62% | 99.62% | 99.62% |
| AUC | 95.86% | 99.25% | 99.70% | 99.77% | 29.25% | 26.84% | 27.59% | 28.35% |
| Inverse AUC | 21.43% | 19.17% | 16.32% | 14.21% | 99.92% | 99.92% | 99.92% | 99.92% |
| Correlation | 100.00% | 100.00% | 99.92% | 99.92% | 99.92% | 99.92% | 99.92% | 99.92% |
| Correlation(T&R) | 97.29% | 99.32% | 99.85% | 99.77% | 46.02% | 52.48% | 54.51% | 55.19% |
| Spearman Correlation | 53.91% | 54.14% | 54.59% | 55.64% | 37.07% | 34.96% | 35.86% | 35.04% |
| Spearman Correlation(T&R) | 93.76% | 96.39% | 97.67% | 98.65% | 50.90% | 57.97% | 51.95% | 53.83% |
| Cosine | 100.00% | 100.00% | 100.00% | 100.00% | 99.85% | 99.85% | 99.85% | 99.85% |
| WPMI | 97.07% | 100.00% | 100.00% | 100.00% | 99.70% | 99.77% | 99.77% | 99.85% |
| MAD | 50.75% | 48.95% | 49.40% | 39.17% | 99.70% | 99.70% | 99.70% | 99.70% |
| AUPRC | 99.40% | 99.32% | 99.32% | 99.32% | 99.62% | 99.62% | 99.62% | 99.62% |
| Inverse AUPRC | 100.00% | 100.00% | 100.00% | 100.00% | 99.62% | 99.62% | 99.62% | 99.62% |

*Table E.5.* A study measuring whether our test results are sensitive to sampling $c_t^{\pm}$ only once per neuron. We see this randomness has no effect on our results in terms of which metrics pass and do not pass the test, likely because the results are already averaged over hundreds or thousands of neurons.

## E.6. Combination Metrics

In addition to individual metrics we studied in the main paper, we also explore whether combinations of different metrics could serve as good evaluation metrics. This investigation is inspired by the success of F1-score, which is the harmonic mean of the *Recall* and *Precision* metrics.

$$s_{F1} = \frac{2}{1/s_{\text{Recall}} + 1/s_{\text{Precision}}} = \frac{2 s_{\text{Recall}} s_{\text{Precision}}}{s_{\text{Recall}} + s_{\text{Precision}}} \tag{E.2}$$

In this section we investigated using the harmonic mean of other metrics as our metric. The missing and extra labels test results are reported in Tables E.6, E.7 and E.8. We can see multiple combinations perform well and now pass the theoretical missing/extra labels tests. In general, our initial results indicate that combining a metric that passes the extra labels test with a metric that passes the missing labels test will pass both tests most of the time. Conversely, combining two metrics that fail the same test, such as Recall and AUC will still fail that test. In table E.6, we also measure the AUPRC (Sec 5) performance of these new metrics on a subset of our settings, and show that many combination metrics are competitive with top standard metrics. In particular the harmonic mean of Balanced Acc and Inverse Balanced Acc performed well. Overall this shows the promise of using novel combinations of concepts, but a more throughout study is needed to confirm their reliability.

| | Theoretical Missing Labels Test, Decrease Acc | | | | | |
|---|---|---|---|---|---|---|
| **Activation Frequency $\gamma$:** | 0.499 | 0.1 | 0.01 | 0.001 | 0.0001 | Pass |
| **AUC + Inverse AUC** | 100.00% | 100.00% | 100.00% | 100.00% | 100.00% | ✓ |
| **Balanced Acc + Inverse Balanced Acc** | 100.00% | 100.00% | 100.00% | 100.00% | 100.00% | ✓ |
| **Recall + AUC** | 100.00% | 100.00% | 100.00% | 100.00% | 100.00% | ✓ |
| **Recall + Inverse AUC** | 100.00% | 100.00% | 100.00% | 100.00% | 100.00% | ✓ |
| **Precision + Balanced Acc** | 100.00% | 100.00% | 100.00% | 100.00% | 100.00% | ✓ |
| **Precision + Inverse Balanced Acc** | 100.00% | 100.00% | 100.00% | 0.00% | 0.00% | ✗ |
| **F1-score** | 100.00% | 100.00% | 100.00% | 100.00% | 100.00% | ✓ |

*Table E.6.* Theoretical Missing Labels test on combination metrics.

| | Theoretical Extra Labels Test, Decrease Acc | | | | | |
|---|---|---|---|---|---|---|
| **Activation Frequency $\gamma$:** | 0.499 | 0.1 | 0.01 | 0.001 | 0.0001 | Pass |
| **AUC + Inverse AUC** | 100.00% | 100.00% | 100.00% | 100.00% | 100.00% | ✓ |
| **Balanced Acc + Inverse Balanced Acc** | 100.00% | 100.00% | 100.00% | 100.00% | 100.00% | ✓ |
| **Recall + AUC** | 100.00% | 100.00% | 100.00% | 0.00% | 0.00% | ✗ |
| **Recall + Inverse AUC** | 100.00% | 100.00% | 100.00% | 100.00% | 100.00% | ✓ |
| **Precision + Balanced Acc** | 100.00% | 100.00% | 100.00% | 100.00% | 100.00% | ✓ |
| **Precision + Inverse Balanced Acc** | 100.00% | 100.00% | 100.00% | 100.00% | 100.00% | ✓ |
| **F1-score** | 100.00% | 100.00% | 100.00% | 100.00% | 100.00% | ✓ |

*Table E.7.* Theoretical Extra Labels Test for combination metrics.

| | Setting 1: ViT-B-16 final layer neuron | | Setting 2: Resnet-50 layer4 neurons | |
| | Missing Labels Test | Extra Labels Test | Missing Labels Test | Extra Labels Test |
| Combination Metric | Decrease acc | Decrease acc | Decrease acc | Decrease acc |
|---|---|---|---|---|
| AUC + Inverse AUC | 95.64% | 99.77% | 89.05% | 99.90% |
| Balanced Acc + Inverse Balanced Acc | 99.85% | 99.92% | 90.75% | 100.00% |
| Recall + AUC | 96.99% | 19.77% | 100.00% | 17.32% |
| Recall + Inverse AUC | 96.92% | 99.92% | 100.00% | 87.00% |
| Precision + Balanced Acc | 91.80% | 99.70% | 64.39% | 100.00% |
| Precision + Inverse Balanced Acc | 48.87% | 99.92% | 47.38% | 100.00% |
| F1-score | 95.86% | 99.70% | 100.00% | 100.00% |

*Table E.8.* Experimental Missing and Extra Labels Test with new combination metrics.

| | Setting 3: gt $c_t$ | | Setting 4: SigLIP $c_t$ | |
| | ViT-B-16(ImageNet) | | ViT-B-16(ImageNet) | |
| New metrics: | AUPRC | Rank | AUPRC | Rank |
|---|---|---|---|---|
| AUC + Inverse AUC | 0.8977 | 3 | 0.6551 | 10 |
| Balanced Acc + Inverse Balanced Acc | 0.8885 | 5 | 0.7041 | 3 |
| Recall + AUC | 0.7081 | 17 | 0.6129 | 15 |
| Recall + Inverse AUC | 0.8892 | 4 | 0.0077 | 22 |
| Precision + Balanced Acc | 0.8679 | 8 | 0.7141 | 2 |
| Precision + Inverse Balanced Acc | 0.8610 | 9 | 0.6468 | 12 |
| **Old metrics:** | | | | |
| Recall | 0.0832 | 22 | 0.5758 | 17 |
| Precision | 0.8592 | 11 | 0.6428 | 13 |
| F1-score/IoU | 0.8759 | 7 | 0.6992 | 5 |
| Accuracy | 0.7858 | 13 | 0.5853 | 16 |
| Balanced Accuracy | 0.7463 | 15 | 0.6158 | 14 |
| Inverse Balanced Acc. | 0.8610 | 9 | 0.6474 | 11 |
| AUC | 0.7139 | 16 | 0.6933 | 7 |
| Inverse AUC | 0.8120 | 12 | 0.5439 | 18 |
| Correlation | **0.9399** | **1** | 0.7027 | 4 |
| Correlation(T&R) | 0.4365 | 18 | 0.4906 | 19 |
| Spearman Correlation | 0.0125 | 23 | 0.0047 | 23 |
| Spearman Correlation(T&R) | 0.1783 | 20 | 0.2395 | 20 |
| Cosine | 0.9037 | 2 | 0.6948 | 6 |
| WPMI | 0.7570 | 14 | 0.6881 | 8 |
| MAD | 0.1866 | 19 | 0.1268 | 21 |
| AUPRC | 0.8839 | 6 | **0.7511** | **1** |
| Inverse AUPRC | 0.1131 | 21 | 0.6688 | 9 |

*Table E.9.* Performance of new combination on the meta-AUPRC evaluation introduced in section 5. We can see some combination metrics such as Balanced Acc + Inverse Balanced Acc perform well.

# F. Detailed results

## F.1. Missing and Extra Labels Test:

**Experimental Setup Details:** We evaluate our experimental results across 8 settings:

- **Setting 1:** Dataset $\mathcal{D}$=ImageNet, ViT-B-16 (Dosovitskiy et al., 2021) trained on imagenet, 1400 final layer neurons(and superclass neurons), Table F.1

- **Setting 2:** Dataset $\mathcal{D}$=ImageNet, ResNet-50 trained on imagenet, 2048 layer4 neurons, Table F.1

- **Setting 3:** Dataset $\mathcal{D}$=Places365, ResNet-18 trained on Places365, 365 final layer neurons, Table F.2

- **Setting 4:** Dataset $\mathcal{D}$=Places365, ResNet-18 trained on Places365, 512 layer4 neurons, Table F.2

- **Setting 5:** Dataset $\mathcal{D}$=CUB200, CBM trained on CUB200, 112 concept neurons, Table F.3

- **Setting 6:** Dataset $\mathcal{D}$=CUB200, CLIP ViT-B-32 image encoder, linear probe trained to detect CUB-concepts, 112 concepts, Table F.3

- **Setting 7:** Dataset $\mathcal{D}$=OpenWebText(subset), GPT-2-small, final(prediction) layer neurons corresponding to 500 most common tokens, Table F.4

- **Setting 8:** Dataset $\mathcal{D}$=OpenWebText(subset), GPT-2-XL, final(prediction) layer neurons corresponding to 500 most common tokens, Table F.4

For all settings we used the ground truth labels from the dataset as $c_t$. The ImageNet (Deng et al., 2009), Places (Zhou et al., 2017) and GPT-2 (Radford et al., 2019) models were pretrained. For CUB-CBM we trained our own model using the code released by (Koh et al., 2020). Our CBM reached 96.75% concept accuracy on the test set which is in line with their reported results. For CLIP, we used the pretrained model from (Radford et al., 2021), and then learned a linear probe on top of frozen image embeddings to minimize binary cross-entropy loss on the training split of CUB200(Wah et al., 2011), with early stopping using validation data. Our linear probe reached 89.76% concept accuracy. The CUB dataset is a small bird species classification dataset that contains detailed annotations for lower level concepts, such as wing color. Following (Koh et al., 2020), we only used the 112 concepts that are present on at least 5% of the inputs and our CLIP linear probe was trained to predict these concepts, not the final class of inputs.

For the final layer neurons as well as CUB neurons we let "correct" concept $t_k$ be the ground truth concept for that neuron. We choose the hyperparameter $\alpha$ that maximizes AUPRC(Sec 5) performance on validation neurons, and run the tests on test neurons. For all evaluations we used neuron activations after the activation function (i.e. softmax/sigmoid).

For the Language Model evaluations we used a subset of 204,800 tokens from OpenWebText (Gokaslan et al., 2019) as $\mathcal{D}$. We evaluated final layer neurons that directly predict next-token, so their explanation was a specific token, which allowed us the extract ground truth $c_t$ directly from the text. We focused on the neurons predicting 500 most common tokens to reduce computational cost and to make sure we focus on concepts where there are enough positive labels available for statistical significance.

For layer4(after avg pool) neurons we defined the "correct" concept $t_k$ as the concept that maximizes IoU with $\alpha = 0.005$ similar to (Bau et al., 2017), using the class(and superclass) labels of the dataset as $c_t$. For these layers we fixed $\alpha = 0.005$ for all metrics as that was used to determine the "ground truth".

Interestingly, we find that more methods pass the tests on the CUB dataset than the other datasets we looked at, but the trends in terms of which metrics perform worse are still similar. We believe this is caused by data imbalance, as the concepts in CUB are relatively balanced (following (Koh et al., 2020) we only keep concepts that are present on at least 5% of the inputs), while for example ImageNet classes are much more imbalanced (each class is positive on 0.1% of the inputs). This is confirmed by our theoretical observations in Section C, which show that poor metrics are much more likely to fail the test when the concepts are imbalanced.

| | Setting 1: ViT-B-16 final layer neurons | | Setting 2: Resnet-50 layer4 neurons | |
|---|---|---|---|---|
| | **Missing Labels Test** | **Extra Labels Test** | **Missing Labels Test** | **Extra Labels Test** |
| **Metric** | **Decrease Acc** | **Decrease Acc** | **Decrease Acc** | **Decrease Acc** |
| **Recall** | 95.94% | 0.00% | 100.00% | 0.00% |
| **Precision** | 46.09% | 99.62% | 47.33% | 100.00% |
| **F1-score** | 95.64% | 99.70% | 100.00% | 100.00% |
| **IoU** | 95.34% | 99.62% | 100.00% | 100.00% |
| **Accuracy** | 0.00% | 63.08% | 0.00% | 71.07% |
| **Balanced Acc.** | 95.86% | 23.98% | 100.00% | 21.12% |
| **Inverse Balanced Acc.** | 45.56% | 99.62% | 46.35% | 100.00% |
| **AUC** | 95.34% | 30.23% | 87.41% | 49.02% |
| **Inverse AUC** | 21.35% | 99.92% | 46.61% | 100.00% |
| **Correlation** | 99.92% | 99.92% | 100.00% | 100.00% |
| **Correlation(T&R)** | 98.12% | 44.96% | 85.05% | 50.51% |
| **Spearman Correlation** | 53.91% | 38.05% | 56.89% | 38.08% |
| **Spearman Correlation(T&R)** | 93.61% | 50.75% | 67.47% | 48.72% |
| **Cosine** | 100.00% | 99.85% | 100.00% | 97.84% |
| **WPMI** | 95.94% | 99.70% | 86.64% | 100.00% |
| **MAD** | 49.47% | 99.70% | 46.30% | 100.00% |
| **AUPRC** | 99.25% | 99.62% | 99.23% | 100.00% |
| **Inverse AUPRC** | 100.00% | 99.62% | 98.05% | 98.05% |

*Table F.1.* Detailed results of Missing/Extra Labels test on Setting 1 and 2.

| | Setting 3: RN18(Places365) final layer | | Setting 4: RN18(Places365) layer4 | |
|---|---|---|---|---|
| | **Missing Labels Test** | **Extra Labels Test** | **Missing Labels Test** | **Extra Labels Test** |
| **Metric** | **Decrease Acc** | **Decrease Acc** | **Decrease Acc** | **Decrease Acc** |
| **Recall** | 95.68% | 0.00% | 100.00% | 0.00% |
| **Precision** | 48.13% | 98.85% | 45.38% | 100.00% |
| **F1-score** | 56.77% | 98.85% | 99.79% | 100.00% |
| **IoU** | 56.77% | 98.85% | 99.79% | 100.00% |
| **Accuracy** | 0.00% | 100.00% | 0.00% | 100.00% |
| **Balanced Acc.** | 95.68% | 99.42% | 100.00% | 65.71% |
| **Inverse Balanced Acc.** | 44.67% | 98.85% | 45.17% | 100.00% |
| **AUC** | 99.14% | 59.37% | 87.89% | 51.33% |
| **Inverse AUC** | 41.50% | 100.00% | 50.31% | 100.00% |
| **Correlation** | 99.14% | 99.42% | 100.00% | 100.00% |
| **Correlation(T&R)** | 95.10% | 48.70% | 83.16% | 51.33% |
| **Spearman Correlation** | 60.23% | 37.46% | 64.07% | 44.76% |
| **Spearman Correlation(T&R)** | 89.91% | 51.59% | 70.43% | 50.51% |
| **Cosine** | 99.42% | 99.42% | 100.00% | 99.79% |
| **WPMI** | 95.68% | 0.00% | 91.17% | 100.00% |
| **MAD** | 46.97% | 99.71% | 45.38% | 100.00% |
| **AUPRC** | 84.44% | 97.98% | 97.13% | 100.00% |
| **Inverse AUPRC** | 100.00% | 99.71% | 99.18% | 100.00% |

*Table F.2.* Detailed results of Missing/Extra Labels test on Setting 3 and 4.

| | Setting 5: CUB200 CBM neurons | | Setting 6: CUB200 Linear Probe | |
| | Missing Labels Test | Extra Labels Test | Missing Labels Test | Extra Labels Test |
| Metric | Decrease Acc | Decrease Acc | Decrease Acc | Decrease Acc |
|---|---|---|---|---|
| Recall | 100.00% | 0.00% | 100.00% | 0.00% |
| Precision | 38.32% | 100.00% | 43.93% | 100.00% |
| F1-score | 100.00% | 100.00% | 100.00% | 100.00% |
| IoU | 100.00% | 100.00% | 100.00% | 100.00% |
| Accuracy | 86.92% | 100.00% | 100.00% | 95.33% |
| Balanced Acc. | 100.00% | 100.00% | 100.00% | 100.00% |
| Inverse Balanced Acc. | 100.00% | 100.00% | 98.13% | 100.00% |
| AUC | 100.00% | 100.00% | 100.00% | 90.65% |
| Inverse AUC | 100.00% | 100.00% | 100.00% | 100.00% |
| Correlation | 100.00% | 100.00% | 100.00% | 100.00% |
| Correlation(T&R) | 99.07% | 93.46% | 98.13% | 94.39% |
| Spearman Correlation | 100.00% | 90.65% | 100.00% | 87.85% |
| Spearman Correlation(T&R) | 93.46% | 89.72% | 95.33% | 85.05% |
| Cosine | 100.00% | 100.00% | 100.00% | 97.20% |
| WPMI | 100.00% | 91.59% | 100.00% | 79.44% |
| MAD | 95.33% | 99.07% | 96.26% | 96.26% |
| AUPRC | 100.00% | 100.00% | 100.00% | 100.00% |
| Inverse AUPRC | 100.00% | 93.46% | 100.00% | 73.83% |

*Table F.3.* Detailed results of Missing/Extra Labels test on Setting 5 and 6.

| | Setting 7: GPT-2-small final layer | | Setting 8: GPT-2-XL final layer | |
| | Missing Labels Test | Extra Labels Test | Missing Labels Test | Extra Labels Test |
| Metric | Decrease Acc | Decrease Acc | Decrease Acc | Decrease Acc |
|---|---|---|---|---|
| Recall | 98.11% | 0.00% | 99.58% | 0.00% |
| Precision | 47.16% | 100.00% | 49.47% | 100.00% |
| F1-score | 99.37% | 100.00% | 97.89% | 100.00% |
| IoU | 99.16% | 100.00% | 97.89% | 100.00% |
| Accuracy | 1.47% | 16.63% | 1.89% | 16.84% |
| Balanced Acc. | 98.11% | 9.68% | 99.58% | 9.47% |
| Inverse Balanced Acc. | 45.47% | 100.00% | 48.84% | 99.16% |
| AUC | 93.89% | 43.79% | 96.00% | 49.05% |
| Inverse AUC | 34.11% | 100.00% | 28.63% | 100.00% |
| Correlation | 97.47% | 100.00% | 98.74% | 100.00% |
| Correlation(T&R) | 69.26% | 48.00% | 74.74% | 50.74% |
| Spearman Correlation | 38.53% | 26.95% | 38.74% | 29.89% |
| Spearman Correlation(T&R) | 65.05% | 52.21% | 65.05% | 49.89% |
| Cosine | 97.47% | 100.00% | 98.74% | 100.00% |
| WPMI | 98.11% | 0.00% | 99.58% | 0.00% |
| MAD | 49.68% | 100.00% | 49.05% | 100.00% |
| AUPRC | 91.16% | 98.32% | 93.68% | 99.79% |
| Inverse AUPRC | 97.68% | 100.00% | 98.32% | 100.00% |

*Table F.4.* Detailed results of Missing/Extra Labels test on Setting 7 and 8.

## F.2. Meta-AUPRC evaluation:

**Experimental Setup Details:** We evaluate our meta-AUPRC test described in Section 5 on 10 different settings:

- **Setting 1:** Dataset $\mathcal{D}$=ImageNet, ViT-B-16(ImageNet), 1000 final layer neurons, ground truth $c_t$

- **Setting 2:** Dataset $\mathcal{D}$=ImageNet, ViT-B-16(ImageNet), 1000 final layer neurons, SigLIP $c_t$

- **Setting 3:** Dataset $\mathcal{D}$=ImageNet, ViT-B-16(ImageNet), 1400 final layer neurons+superclass neurons, ground truth $c_t$

- **Setting 4:** Dataset $\mathcal{D}$=ImageNet, ViT-B-16(ImageNet), 1400 final layer neurons+superclass neurons, SigLIP $c_t$

- **Setting 5:** Dataset $\mathcal{D}$=Places365, ResNet-18(Places365), 365 final layer neurons, ground truth $c_t$

- **Setting 6:** Dataset $\mathcal{D}$=Places365, ResNet-18(Places365), 365 final layer neurons, SigLIP $c_t$

- **Setting 7:** Dataset $\mathcal{D}$=CUB200, CBM trained on CUB200, 112 concept neurons, ground truth $c_t$

- **Setting 8:** Dataset $\mathcal{D}$=CUB200, CLIP ViT-B-32 image encoder, linear probe trained to detect 112 CUB-concepts, ground truth $c_t$

- **Setting 9:** Dataset $\mathcal{D}$=OpenWebText(subset), GPT-2-small, final(prediction) layer neurons corresponding to 500 most common tokens.

- **Setting 10:** Dataset $\mathcal{D}$=OpenWebText(subset), GPT-2-XL, final(prediction) layer neurons corresponding to 500 most common tokens.

SigLIP $c_t$ indicates we used Pseudo-labels generated from SigLIP (ViT-SO400M-14-SigLIP-384) (Zhai et al., 2023) as done by (Oikarinen & Weng, 2024). For all metrics and evaluations we choose hyperparameters such as $\alpha$ by finding the one with best performance on validation neurons (random subset of 5% of the neurons), and use those hyperparameters to evaluate on test neurons.

Tables F.5, F.6 and F.7 show the detailed results of our AUPRC evaluation experiment. While there is some differences on which metrics perform well on different setups, the overall trends are quite consistent, with Correlation and Cosine outperforming others on almost all settings, and AUPRC being the third best. Spearman Correlation also consistently performed the worst.

| Metric | Setting 1: gt $c_t$ original $K$ and $C$ AUPRC | Rank | Setting 2: SigLIP $c_t$ original $K$ and $C$ AUPRC | Rank | Setting 3: gt $c_t$ original+superclass $K$ and $C$ AUPRC | Rank | Setting 4: SigLIP $c_t$ original+superclass $K$ and $C$ AUPRC | Rank |
|---|---|---|---|---|---|---|---|---|
| Recall | 0.9989 | 6 | 0.8638 | 12 | 0.0832 | 16 | 0.5758 | 12 |
| Precision | 0.9989 | 6 | 0.9640 | 6 | 0.8592 | 6 | 0.6428 | 9 |
| F1-score/IoU | 0.9989 | 6 | 0.9477 | 7 | 0.8759 | 4 | 0.6992 | 3 |
| Accuracy | 0.9989 | 6 | 0.8497 | 13 | 0.7858 | 8 | 0.5853 | 11 |
| Balanced Acc. | 0.9989 | 6 | 0.9032 | 11 | 0.7463 | 10 | 0.6158 | 10 |
| Inverse Balanced Acc. | 0.9989 | 6 | 0.9664 | 5 | 0.8610 | 5 | 0.6474 | 8 |
| AUC | 0.9974 | 14 | 0.9438 | 8 | 0.7139 | 11 | 0.6933 | 5 |
| Inverse AUC | 0.9415 | 15 | 0.8452 | 14 | 0.8120 | 7 | 0.5439 | 13 |
| Correlation | 0.9998 | 2 | 0.9739 | 2 | **0.9399** | **1** | 0.7027 | 2 |
| Correlation(T&R) | 0.9993 | 5 | 0.7257 | 15 | 0.4365 | 12 | 0.4906 | 14 |
| Spearman Correlation | 0.0023 | 17 | 0.0213 | 17 | 0.0125 | 17 | 0.0047 | 17 |
| Spearman Correlation(T&R) | 0.6533 | 16 | 0.4100 | 16 | 0.1783 | 14 | 0.2395 | 15 |
| Cosine | 0.9998 | 2 | 0.9733 | 3 | 0.9037 | 2 | 0.6948 | 4 |
| WPMI | 0.9989 | 6 | 0.9211 | 9 | 0.7570 | 9 | 0.6881 | 6 |
| MAD | 0.9994 | 4 | 0.9716 | 4 | 0.1866 | 13 | 0.1268 | 16 |
| AUPRC | 0.9989 | 6 | **0.9866** | **1** | 0.8839 | 3 | **0.7511** | **1** |
| Inverse AUPRC | **0.9999** | **1** | 0.9203 | 10 | 0.1131 | 15 | 0.6688 | 7 |

*Table F.5.* Meta-AUPRC evaluation results on ImageNet Settings(1-4).

| Metric | Setting 5: gt $c_t$ Resnet-18(Places) AUPRC | Rank | Setting 6: SigLIP $c_t$ Resnet-18(Places) AUPRC | Rank | Setting 7: gt $c_t$ CBM(CUB200) AUPRC | Rank | Setting 8: gt $c_t$ Linear Probe(CUB200) AUPRC | Rank |
|---|---|---|---|---|---|---|---|---|
| Recall | 0.9639 | 12 | 0.7623 | 9 | 0.3958 | 17 | 0.1078 | 17 |
| Precision | 0.9640 | 6 | 0.7771 | 8 | 0.6803 | 9 | 0.2095 | 12 |
| F1-score/IoU | 0.9640 | 6 | 0.7916 | 6 | 0.6386 | 12 | 0.2792 | 7 |
| Accuracy | 0.9640 | 6 | 0.6490 | 14 | 0.5853 | 13 | 0.2239 | 10 |
| Balanced Acc. | 0.9640 | 6 | 0.7340 | 11 | 0.7165 | 7 | 0.3186 | 4 |
| Inverse Balanced Acc. | 0.9640 | 6 | 0.7813 | 7 | 0.6628 | 11 | 0.2608 | 8 |
| AUC | 0.9418 | 14 | 0.7205 | 12 | 0.7045 | 8 | 0.2031 | 14 |
| Inverse AUC | 0.8927 | 15 | 0.6877 | 13 | 0.8956 | 2 | 0.3745 | 3 |
| Correlation | **0.9693** | **1** | 0.8326 | 2 | 0.8883 | 3 | **0.4683** | **1** |
| Correlation(T&R) | 0.9662 | 4 | 0.5326 | 15 | 0.7170 | 6 | 0.2976 | 6 |
| Spearman Correlation | 0.0213 | 17 | 0.1214 | 17 | 0.4233 | 15 | 0.2185 | 11 |
| Spearman Correlation(T&R) | 0.7424 | 16 | 0.4005 | 16 | 0.4243 | 14 | 0.1607 | 15 |
| Cosine | **0.9693** | **1** | 0.8304 | 4 | **0.8985** | **1** | 0.4067 | 2 |
| WPMI | 0.9655 | 5 | 0.7539 | 10 | 0.7323 | 5 | 0.2089 | 13 |
| MAD | 0.9590 | 13 | 0.8321 | 3 | 0.7760 | 4 | 0.2432 | 9 |
| AUPRC | 0.9640 | 6 | **0.8669** | **1** | 0.6712 | 10 | 0.2981 | 5 |
| Inverse AUPRC | 0.9678 | 3 | 0.8006 | 5 | 0.3975 | 16 | 0.1302 | 16 |

*Table F.6.* Meta-AUPRC evaluation results on Places and CUB models Settings(5-8).

| | Setting 9: gt $c_t$ | | Setting 10: gt $c_t$ | |
|---|---|---|---|---|
| | GPT2-small | | GPT2-XL | |
| Metric | AUPRC | Rank | AUPRC | Rank |
| Recall | 0.9754 | 6 | 0.9951 | 6 |
| Precision | 0.9655 | 8 | 0.9780 | 9 |
| F1-score/IoU | 0.9569 | 9 | 0.9875 | 7 |
| Accuracy | 0.7395 | 13 | 0.8334 | 14 |
| Balanced Acc. | 0.9842 | 4 | 0.9973 | 4 |
| Inverse Balanced Acc. | 0.9658 | 7 | 0.9782 | 8 |
| AUC | 0.8139 | 12 | 0.9194 | 12 |
| Inverse AUC | 0.7282 | 14 | 0.8481 | 13 |
| Correlation | **0.9918** | **1** | **0.9986** | **1** |
| Correlation(T&R) | 0.6705 | 15 | 0.7695 | 15 |
| Spearman Correlation | 0.0137 | 17 | 0.0142 | 17 |
| Spearman Correlation(T&R) | 0.0917 | 16 | 0.1168 | 16 |
| Cosine | 0.9912 | 2 | 0.9985 | 2 |
| WPMI | 0.9773 | 5 | 0.9958 | 5 |
| MAD | 0.9014 | 11 | 0.9557 | 11 |
| AUPRC | 0.9880 | 3 | 0.9951 | 6 |
| Inverse AUPRC | 0.9314 | 10 | 0.9780 | 9 |

*Table F.7.* Meta-AUPRC evaluation results on Language Models Settings(9-10).

