# OpenReview forum: "Evaluating Neuron Explanations: A Unified Framework with Sanity Checks"
_ICML.cc/2025/Conference — ICML 2025 poster_

### Official Review · Reviewer_UBi6 · 2025-03-04

**Overall Recommendation:** 4

**Summary:**

This paper introduces a systematic framework designed for evaluating neural explanations. The core of this framework lies in its ability to quantitatively measure how well a given explanation aligns with neuron behavior across different samples.

The framework operates on a few key components. First, for any explanation *t* (which could be textual, visual, or any other format), it defines $[c_t]_i$ as a measure that indicates the presence of concept *t* in sample *i*. This presence is potentially determined through methods like crowdsourcing. Second, for a neuron *k*, $[a_k]_i$ represents its activation value on sample *i*. The alignment between $[a_k]_i$ and $[c_t]_i$ is then framed as a binary classification task. In this task, perfect matching suggests that explanation *t* accurately describes neuron *k*'s behavior. The framework also formalizes various evaluation metrics from previous research, such as measuring the **Recall** or **IoU (Intersection over Union)** between $c_t$ and $a_k$.

Furthermore, the paper introduces two meta-evaluation approaches aimed at assessing the reliability of these evaluation metrics. The first is the **Label Manipulation Test**. This test involves artificially setting $[c_t]_i$ to 0 for samples that originally contained the concept. Reliable metrics should then show decreased performance compared to the initial perfect score. Conversely, when $[c_t]_i$ is set to 1 for samples that lacked the concept, scores should similarly decrease from their initial perfect score. The paper conducts these tests both theoretically and empirically for various metrics.

The second approach is the **Neurons with Known Concepts Test**, which uses the final layer neurons of classifiers with known explanations as ground truth. The framework computes scores for all neuron-concept pairs, including both correct pairs and random pairings. The expectation is that reliable evaluation metrics should consistently yield higher scores for correct neuron-concept pairs when compared to incorrect ones.

The paper conclude by suggesting some metrices as most reliable options in their framework.

## update after rebuttal
I kept my score as autors addressed the issues I raised.

**Claims And Evidence:**

Yes

**Essential References Not Discussed:**

No

**Experimental Designs Or Analyses:**

The experiment on Meta-evaluation **Label manipulation Test** is valid. However its contribution to the paper claims is very minimal. Their Theoretical analysis can fully support those claims without the need of the expeiments.

**Methods And Evaluation Criteria:**

Yes

**Other Comments Or Suggestions:**

1. The addition of examples in Appendix F (e.g., a sample dataset input, a concept present in that input, corresponding $c_t$ and $a_k$ values for a neuron) would enhance understanding.

2. Also, there appears to be a writing issue on Line 1636 in the appendix that needs to be addressed.

**Other Strengths And Weaknesses:**

# Strengths

1.  The paper addresses a very crucial problem. Evaluation in machine learning interpretability is currently inconsistent, with each work often employing unique, new evaluation methods. As a result comparison of different methods is very hard.
2.  The paper's formalization is simple yet comprehensive, effectively encompassing the evaluation setups used in many other works.
3.  The proposed meta-evaluation metrics are both logical and a valuable contribution to the field.
4. The paper is very well written and easy to follow.

# Weaknesses

1.  The paper neglects to discuss the labeling costs associated with each metric, a significant consideration. For example, while the paper criticizes the use of **recall**, it's important to recognize that **recall** can reduce labeling costs because it only requires concept labels ($[c_t]_i$) for samples that highly activate a neuron. Given that calculating $c_t$ typically involves crowdsourcing or LLMs (the most expensive part of the evaluation process), **recall** offers a practical advantage. Furthermore, crowdsourcing labels for $c_t$ across *all* samples, where the label is predominantly 0 (due to concepts appearing in only a small percentage of samples), can lead to lower-quality labels due to extreme data imbalance (labelers may become inattentive and default to labeling most samples as 0).
2.  The paper's meta-evaluation using **Neurons with Known Concepts** lacks proper citation. Soltani Moakhar et al. [1] present a very similar method for evaluating interpretability, employing the final layer of a ResNet (where neuron/concept pairs are known) for evaluation purposes. While this paper uses it for meta-evaluation (distinguishing it somewhat), it's still a relevant prior art to acknowledge.
3.  The paper's novelty is limited. While not necessarily a weakness, a completely novel and unintuitive evaluation metric might face resistance to adoption within the field.

[1]: Soltani Moakhar, A., Iofinova, E., Frantar, & Alistarh, D. SPADE: Sparsity-Guided Debugging for Deep Neural Networks. ICML 2024.

**Questions For Authors:**

.

**Relation To Broader Scientific Literature:**

This paper formalized previous evaluation metrices in an unified framework.

**Theoretical Claims:**

I checked the theoretical analysis in section D of appendix which is related to **Label Manipulation Test** Meta-evaluation. No issue found.

---

> ### Author Rebuttal · Authors · 2025-04-01
>
> Thank you for the review and positive feedback! Your summary is very well written and highlights a strong understanding of our work. We would like to address your concerns below:
>
> **Weakness 1 - Labeling Cost**
>
> This is a great point. It is true that a significant reason for the use of recall is that it is cheaper to evaluate as we only need to only evaluate $c_{ti}$ on highly activating inputs, i.e. where $a_{ki} = 1$. In contrast, evaluating most other metrics requires knowledge of the full $c_t$. One way to address this issue is to use a combination of metrics, for example evaluating F1-score by combining a crowdsourced evaluation of Recall with a generative model based evaluation of Precision as we suggest in Line 897 of the Appendix.
>
> As you mentioned, crowdsourced evaluation of the entire $c_t$ will likely give noisy results as most inputs do not contain the concept. To avoid this, another approach is to oversample highly activating inputs similarly to Top-and-random sampling. This can be done without failing the extra labels test if the proper importance sampling correction is applied to correct for the bias introduced by the sampling. However, the effective sampling/user study design for non-recall metrics is a rather large and complex topic that does not fit within the scope of the current paper, but it is something we are actively looking into and we are confident that there are ways to evaluate the other metrics with a cost not much higher than recall evaluation. We will include discussion of this under limitations/future work.
>
> **Weakness 2 - Missing citation**
>
> Thank you for pointing this out. We will cite [1] as well as other papers[2, 3] conducting similar studies based on neurons with known concepts as suggested by Reviewer 7Dju.
>
> [1] Soltani Moakhar, A., Iofinova, E., Frantar, & Alistarh, D. SPADE: Sparsity-Guided Debugging for Deep Neural Networks. ICML 2024.
>
> [2] Schwettmann et al, "FIND: A Function Description Benchmark for Evaluating Interpretability Methods", 2023.
>
> [3] Shaham et al, "A Multimodal Automated Interpretability Agent", 2025.
>
> **Weakness 3 - Limited novelty i.e. not introducing new metrics**
>
> Similar to how you acknowledge this is as “not necessarily a weakness”, we do not think this is a weakness but a conscious decision as we thought it would be a more valuable contribution to the field to rigorously analyze existing and standard statistical metrics instead of focusing on creating new ad-hoc metrics. This way our contributions are more clearly defined, and designing metrics particularly to do well on our tests risks overfitting them to these specific tasks, or worse tuning the tests themselves in a way that our new metrics would do well.
>
> **Other comments or suggestions 1- Examples**
>
> We have included an example with dataset inputs and concepts values Appendix Figure B.1., to improve clarity similar to your suggestion. Please let us know if you would find additional examples useful or have any additional suggestions.
>
> **Other comments or suggestions 2 - Typo**
>
> Thank you for pointing this out, we have corrected the typo in Line 1636 to: “… we defined the “correct” concept $t_k$ as the …”
>
> **Additional Experiments:**
>
> In response to other reviewer's comments, we have conducted additional experiments such as evaluating novel combinations of metrics, showing the robustness of our results to the choice of $\epsilon$, and showcasing that our results using a random subset as $c_t^-$ are similar to using a real semantic subset. We have also included a new figure showcasing the Missing and Extra Labels tests. These results are available at https://drive.google.com/file/d/1OHMxyMW1KVIzxd2Rd_Hx34qIjVecUVmo/view.
>
> **Summary**
>
> We are happy to hear that you find our work is addressing a crucial problem and produces logical and valuable contributions to help solve it. We believe we have addressed you main remaining concerns regarding labeling cost and missing citations and will include a discussion of these in the revised manuscript. Please let us know if you have any remaining questions and we would be happy to discuss them further.

---

> > ### Comment · Reviewer_UBi6 · 2025-04-05
> >
> > Thank you for the rebuttal. All of my concerns have been addressed. As a suggestion, I think it would be helpful to briefly mention the labeling cost issue of non-recall metrices in the conclusion when recommending those metrices. I think the paper is a valueable addition to the the interpretability comunity and I hence keep my score.

---

> > > ### Author Response · Authors · 2025-04-06
> > >
> > > Thanks again for the response and your thoughtful and positive comments! We will include the labeling cost in the conclusion of the revision.

---

### Official Review · Reviewer_3aXR · 2025-03-12

**Overall Recommendation:** 4

**Summary:**

This paper proposes NeuronEval for the meta-evaluation of input-based explanation metrics. Given a textual explanation of an input (and resultant activation), a variety of metrics have been proposed to evaluate how faithfully the description describes the neuron (or “any scalar function of network inputs”). NeuronEval unifies 19 existing explanation evaluation methods (and 18 different metrics) under 1 framework, and characterises differences between them by varying the metric, the source of the concept vector, granularity of activation vectors, and probing dataset. Motivated by the intuition that a reliable metric should be “minimal” (specific) and “exact” (not too general), they propose 2 necessary (but not sufficient) sanity checks: the missing labels test and extra labels test. They find that most metrics fail these tests (lack in sensitivity and fidelity); point out which metrics reliably pass the tests; provide guidelines on reliability desiderata for future metrics in explanation evaluation.

**Claims And Evidence:**

The claims in this work are that meta-evaluation (of XAI evaluation metrics) is necessary to compare and contrast their reliability, to guide the development and usage of more reliable metrics. Such claims are supported by the introduction and application of NeuronEval, a framework which assesses the reliability, sensitivity and robustness of 18 metrics (across 19 methods) through sanity checks of missing and extra labels. The authors present evidence that the majority of metrics fail or perform suboptimally on one/both tests, which speaks to the importance of this investigation.

**Essential References Not Discussed:**

This paper could more deeply with previous work in XAI robustness analysis – RISE [1], Sobol [2], ROAR [3], MuFidelity [4] – which have exhibited similar intuitions regarding the desired sensitivity of XAI methods to perturbations.

[1] Vitali Petsiuk, Abir Das, and Kate Saenko. RISE: randomized input sampling for explanation of black-box models. In BMVC, pp. 151. BMVA Press, 2018. URL http://bmvc2018.org/ contents/papers/1064.pdf.
[2] Thomas Fel, Rémi Cadène, Mathieu Chalvidal, Matthieu Cord, David Vigouroux, and Thomas Serre. Look at the variance! efficient black-box explanations with sobol-based sensitivity analysis. Advances in neural information processing systems, 34:26005–26014, 2021.
[3] Sara Hooker, Dumitru Erhan, Pieter-Jan Kindermans, and Been Kim. A benchmark for interpretability methods in deep neural networks. NeurIPS, 32, 2019.
[4] Umang Bhatt, Adrian Weller, and José MF Moura. Evaluating and aggregating feature-based model explanations. In Proceedings of the Twenty-Ninth International Conference on International Joint Conferences on Artificial Intelligence, pp. 3016–3022, 2021.

**Experimental Designs Or Analyses:**

The experimental details (e.g. specifics of metrics, concept vector, activation vector, probing dataset) for meta-evaluation are expounded upon throughout the paper and supplement. The authors elaborate on how NeuronEval admits different modalities (image and text), network units (e.g. neuron, channel, scalar function of inputs, SAE features, CBM neurons, linear probes), different explanation methods and metrics – I find their descriptions clear and sufficient. They further elaborate on specific setups for the missing/extra labels sanity checks, and discuss 5 possible outcomes and provide hypotheses for failure cases (e.g. concept imbalance, biassed sampling, using generative models in the evaluation pipeline). I find their experimental design reasonable and sound.

**Methods And Evaluation Criteria:**

This paper proposes NeuronEval, a framework which unifies major XAI evaluation metrics via plug-and-play, as well as 2 associated sanity checks for the fidelity and sensitivity of various metrics. The framework is sufficiently general and expressive; the sanity checks are well-motivated and consistent with existing intuition that optimal explanations should be minimal and descriptive of the input-activation pair, and that faithful evaluation metrics (of these explanations) should be sensitive to changes in model response. The proposed framework and sanity checks are applicable for the task of meta-evaluating the faithfulness of XAI metrics.

**Other Comments Or Suggestions:**

N/A

**Other Strengths And Weaknesses:**

In other sections, I have elaborated at length on the positives of this work. Regarding negatives, I find the originality of meta-evaluating XAI methods and metrics based on their sensitivity and faithfulness interesting but not completely novel (see “essential references”). That said, this work is comprehensive, clarifying/unifying and a worthwhile contribution to interpretability research.

**Questions For Authors:**

N/A

**Relation To Broader Scientific Literature:**

This work addresses an important issue in interpretability research: the lack of a structured framework / unified basis to evaluate and compare existing methods. It engages well both with established work in explainable AI (XAI) and newer work in mechanistic interpretability. The framework is quite general and contributes to a unified understanding of XAI: it is compatible with 19 different input-based explanation techniques, 18 metrics; it evaluates explanations of network “units”, from a single neuron, single channel to scalar functions of network inputs, e.g. sparse autoencoder (SAE) features, concept bottleneck model (CBM) neurons, linear probes.

**Theoretical Claims:**

N/A: there are no theoretical claims in this work.

---

> ### Author Rebuttal · Authors · 2025-04-01
>
> Thank you for the detailed review and positive feedback!
>
> **Re: Additional Related Work:**
>
> Thank you for pointing out these references. While these references focus on a very different type of XAI, in particular local input-importance evaluations and as such cannot be applied to our setting, they contain important ideas about reliable evaluation of explanations and we will cite and discuss them in the related work(Appendix B.4) as follows:
>
> The field of input-importance explanations has seen an evolution in the metrics used, with initial focus on finding the features that humans think are important. Later metrics such as deletion and insertion proposed by [1] allow for more principled evaluation of the explanation fidelity, i.e. whether it actually matches what the model in vision models, and [2] extends the deletion metric to natural language settings. [3] proposes the Remove And Retrain (ROAR) framework as an alternative method for evaluating the quality of input-importance explanations by evaluating whether a model retrained on data without the most important pixels can still solve the task. [4] propose additional checks, such as measuring whether an explanation method has high sensitivity, with the intuition that similar inputs should have similar explanations.
>
> Overall, we think that this line of research [1]-[4] does not reduce the novelty of our contribution, as they are focused on a different problem and produce no actionable insight for our setting of global neuron-level explanations. In addition, these papers are mostly focused on introducing new evaluation metrics, instead of conducting meta-evaluation between existing metrics which is the focus of our work.
>
> **Additional Experiments:**
>
> In response to other reviewer's comments, we have conducted additional experiments such as evaluating novel combinations of metrics, showing the robustness of our results to the choice of $\epsilon$, and showcasing that our results using a random subset as $c_t^-$ are similar to using a real semantic subset. We have also included a new figure showcasing the Missing and Extra Labels tests. These results are available at https://drive.google.com/file/d/1OHMxyMW1KVIzxd2Rd_Hx34qIjVecUVmo/view.
>
> Please let us know if you have additional questions and we would be happy to discuss them further! If not, we hope you consider updating your score as you have noted our work is comprehensive, clarifying/unifying and a worthwhile contribution to interpretability research with little weaknesses.
>
>
>
> **References:**
>
> [1] Vitali Petsiuk, Abir Das, and Kate Saenko. RISE: randomized input sampling for explanation of black-box models. In BMVC, pp. 151. BMVA Press, 2018. URL http://bmvc2018.org/contents/papers/1064.pdf.
>
> [2] Thomas Fel, Rémi Cadène, Mathieu Chalvidal, Matthieu Cord, David Vigouroux, and Thomas Serre. Look at the variance! efficient black-box explanations with sobol-based sensitivity analysis. Advances in neural information processing systems, 34:26005–26014, 2021.
>
> [3] Sara Hooker, Dumitru Erhan, Pieter-Jan Kindermans, and Been Kim. A benchmark for interpretability methods in deep neural networks. NeurIPS, 32, 2019.
>
> [4] Umang Bhatt, Adrian Weller, and José MF Moura. Evaluating and aggregating feature-based model explanations. In Proceedings of the Twenty-Ninth International Conference on International Joint Conferences on Artificial Intelligence, pp. 3016–3022, 2021.

---

> > ### Comment · Reviewer_3aXR · 2025-04-08
> >
> > Thank you for the response and improvements to the manuscript. In response to comments, the authors have reviewed the suggested references in detail and clarified the difference in problem settings but commonalities in sensitivity-related observations. They also presented new results to confirm the functional robustness of the 2 proposed sanity checks (missing and extra label tests); they further performed a granular ablations study of if/how using different (combinations of) metrics, (types of) subsets, epsilon values, concept activations might exert influence on performance on the tests. I find that this work has been strengthened by the review/rebuttal process and raise my score from a 3 -> 4.

---

> > > ### Author Response · Authors · 2025-04-09
> > >
> > > Thanks again for the insightful review and participation in the discussion! We are happy to hear you find the submission stronger after the rebuttal.

---

### Official Review · Reviewer_7Dju · 2025-03-14

**Overall Recommendation:** 3

**Summary:**

The paper proposes NeuronEval, a unified meta-evaluation formalism for assessing neuron explanation evaluation metrics. It reformulates 19 commonly used metrics under a shared mathematical notation. The authors assess the reliability of these metrics using two diagnostic tests—missing labels and extra labels—and analyze which metrics pass these tests theoretically, empirically, and on neurons with known concepts across different models and modalities. The analysis identifies a subset of metrics that consistently align with neuron-concept correspondence.

**Claims And Evidence:**

Overall, the paper and supplementary materials present a comprehensive theoretical framework, supported by empirical and theoretical evaluations. The unification of evaluation metrics is well-supported. However, claims about metric reliability are weakened by concerns about the design of the sanity tests and fixed parameter choices, which limit the generality of the conclusions. Please see more details below.

**Essential References Not Discussed:**

In Section 5, it would be useful to refer to existing analyses of neurons with ground-truth descriptions, such as [1, 2].

[1] Schwettmann et al, "FIND: A Function Description Benchmark for Evaluating Interpretability Methods", 2023.

[2] Shaham et al, "A Multimodal Automated Interpretability Agent", 2025.

**Experimental Designs Or Analyses:**

Overall, the experimental section is comprehensive, covering both theoretical and empirical analyses across neurons from different modalities. However, some key information is missing:
- The total number of neurons analyzed has not been reported.
- The source of concept labels c_t​ is unspecified—whether human-annotated, model-generated, or otherwise.
- Both omissions limit the ability to assess the validity and reproducibility of the findings

**Methods And Evaluation Criteria:**

The unification of evaluation metrics under NeuronEval is a valuable and timely contribution. However, the Missing and Extra Labels sanity tests rely on simplistic manipulations of concept labels (random removal or addition) that do not reflect realistic explanation failures, such as semantic drift, polysemanticity, or context-specific activations. Moreover, the use of a fixed epsilon threshold (0.001) ignores their differing scales and sensitivities, potentially skewing the results.

**Other Comments Or Suggestions:**

None

**Other Strengths And Weaknesses:**

None

**Questions For Authors:**

- How many neurons were used in total in each of the experiments?
- How was the labels c_t generated in the experiments?
- How sensitive are the results to the choice of epsilon? Have you tested different epsilon values?
- Why model incorrect concepts as random subsets/supersets? Did you consider more semantically plausible alternatives (e.g., polysemanticity, partial concepts)?
- How would your tests handle neurons that correctly activate only for specific aspects of a concept (e.g., "flying bird" vs. "bird")?
- Did you explore cases where non-activating inputs may still contain the concept?
- Did you analyze emperically if some metrics perform better in certain modalities (e.g., vision vs. language), and if so, how do you plan to address these differences?

**Relation To Broader Scientific Literature:**

The paper contributes to the growing literature on mechanistic interpretability and neuron explanation evaluation

**Theoretical Claims:**

- Eq. 9 lacks an expectation over the random subset c^{-}_{t}, which would be statistically more rigorous, as the sanity test result would otherwise depend on a single random choice
- The effect of the epsilon threshold (0.001) is not analyzed, even though it directly affects whether a metric passes or fails the tests. Metrics with small score ranges may unfairly fail.
- Failure modes are limited to Missing and Extra Labels; other important failures like contextual specificity, polysemanticity, compositions of concepts are not explored. Thus, while the breadth of experiments is good, the design omits key real-world neuron-concept dynamics, limiting the validity of conclusions.

---

> ### Author Rebuttal · Authors · 2025-04-01
>
> Thanks for the detailed review! We have conducted extensive additional experiments available at: https://drive.google.com/file/d/1OHMxyMW1KVIzxd2Rd_Hx34qIjVecUVmo/view, see in particular Tab G5-G7 as these experiments were conducted to address your questions.
>
> **1.Realistic failure modes**
>
> We argue that most real world explanation failures are closely connected to one of two failure modes, and that these failure modes are directly captured by our tests:
>  - FM-A) Explanation **too generic**: This means the explanation concept is a superset of the “true” neuron concept, e.g. describing a neuron as “animals” when it only activates on dogs. Our Extra Labels Test captures whether a metric can detect this failure mode.
> - FM-B) Explanation **too specific**: Explanation concept is a subset of real neuron activations, i.e. describing a neuron as “black cat” when it activates on all cats. Our Missing Labels Test captures whether a metric can detect this failure mode.
>
> Also, our idea of a “concept” is very general and includes any text-based description. This means a single concept could be highly specific (e.g. “flying bird”), or a composition of simpler concepts (e.g. “water OR river”). The realistic failure modes you discussed then fit into the above two failure modes:
> - **Polysemanticity:** A popular model of polysemanticity is to model neuron activations as an OR of different concepts. If the explanation only captures one of these concepts, this means the explanation is too specific (FM-B).
> - **Context specific activations:** A context specific neuron activation means the neuron’s “true” concept is a subset of the non-context specific concept. If the explanation is not-context specific, the explanation is too generic(FM-A).
> - **Non activating inputs still contain the concept:** This means that the explanation concept is too generic i.e. a superset of the “true” concept(FM-A).
>
> **2. Random vs semantic sub/supersets**
>
> Thanks for the suggestion! Our missing/extra labels tests are a mathematical model of semantic sub/superclasses that can be run without knowledge of the semantics or relationships between concepts. To test this model, we conducted a version of extra labels test on the ImageNet dataset where instead of randomly sampling $c^+$ we used the smallest superclass of the concept according to WordNet hierarchy, and a version of missing labels test where we used the largest subclass of the concept as $c^-$. As shown in new Table G.5, the results are essentially identical to using random sub/supersets, showcasing our random sub/superset is a good model of real semantic relationships for our purposes.
>
> **3. Importance of Epsilon Choice:**
>
> To clarify, as mentioned on lines 307-309, we normalize each metric so that their values lie in [0,1] before running the tests to ensure fair comparison. To measure the sensitivity of our results to the choice of eps, we ran our tests with different epsilons as shown in Tab G.7. We can see that changing eps by an order of magnitude does not significantly change which metrics pass.
>
> Our theoretical results also suggest our tests are robust to choice of $\epsilon$. As we show in Tab. C.4 & D.4, the score difference of metrics that fail our tests approaches 0 as the concept/neuron activation frequency $\gamma$ approaches 0, while passing metrics retain a nonzero score difference regardless of $\gamma$. This means that for any $\epsilon > 0$ there exists $\gamma > 0$ s.t. metrics like accuracy will fail the tests.
>
> **4. Lack of expectation over $c^{-}_{t}$:**
>
> Great point. We will add an expectation over $c_t^{-}$ in Eq. 10 as this is a more principled definition. While our experimental results only used one sample of $c_t^-$, we conducted a study on how the results change with additional samples as shown in new Tab G.6. We can see averaging over multiple samples has little impact on the results, likely because we already average over a large number of neurons and settings.
>
> **5. Number of neurons**
>
> Thanks for pointing this out. For the theoretical missing/extra labels test, we simulate 1k neurons as mentioned in line 1129. For empirical Missing/Extra labels test, we evaluated a total of 5549 neurons, and for meta-AUPRC we evaluated a total of 2989 unique neurons.
>
> **6. Source of $c_t$**
>
> We would like to clarify we have specified the source of $c_t$ in the manuscript. For missing/extra labels tests, we use ground truth $c_t$, i.e. from a labeled dataset(line 301-302). For meta-AUPRC, we test with both ground truth $c_t$ from a labeled dataset as well as pseudo-labels from SigLIP(App. F.2, lines 1764-1782).
>
> **7. Additional references**
>
> Thanks, we’ll cite and discuss these references.
>
> **8. Performance on different modalities**
>
> As our main contributions are mathematical features of the metric itself and not tied to any particular model or modality, we did not observe significant differences between the modalities.
>
> Please let us know if you have any remaining questions!

---

> > ### Comment · Reviewer_7Dju · 2025-04-04
> >
> > Thank you for the clarifications and for providing the additional results. I encourage the authors to add the following points to their paper:
> > - Discussions about the accordance of missing/extra label cases in real scenarios.
> > - Results of semantic subsets/supersets vs. random ones.
> > - According to table G.6, the results clearly depend on \epsilon. This is an important point that needs to be further addressed in the paper.
> >
> > The additional results addressed my concerns, and I therefore raise my score to 3: weak accept

---

> > > ### Author Response · Authors · 2025-04-05
> > >
> > > Thank you for the response and valuable feedback! We will include the suggested parts in the updated manuscript as they are important and make the submission stronger.
> > >
> > > Regarding $\epsilon$, while the experimental results(G.6) show some change in response to $\epsilon$, we believe this effect is small as for the most part it does not affect which metrics pass or do not pass the test despite 2 orders of magnitude change in $\epsilon$.
> > >
> > > More importantly, theoretically we do not think the $\epsilon$ choice is important. An alternative and perhaps more fundamental definition for our theoretical tests could be as follows: A metric passes the test if $\exists \epsilon > 0$ s.t. $\forall \gamma > 0, \Delta s < -\epsilon$, where $\Delta s$ is calculated with the theoretical missing labels test defined in Appendix C and $\gamma$ is the concept activation frequency. This definition is purely a mathematical property of the metric without any hyperparameters or ties to any setting, and as far as we know matches our theoretical test results of Table C.1 and C.2 that use a fixed epsilon.
> > >
> > > However, we can only prove a metric passes this version of the test if we have an analytical solution, which we have for binary classification metrics in Table D.2, which makes this definition harder to use in practice. We can see for example accuracy fails this test as $\lim_{\gamma \to 0} \Delta s = 0$ so no matter what $\epsilon$ we choose there exists some $\gamma$ s.t. Accuracy will fail the tests. On the other hand, for F1-score it will pass the theoretical tests for any $\epsilon < 1/3$ regardless of $\gamma$. From this we can see that if we decrease $\epsilon$ in the current theoretical test, more metrics would pass, but the same metrics would fail again if we expand Table C.1/C.2 to the right by including smaller $\gamma$ values, while current passing metrics would not fail regardless of how many additional $\gamma$ values we test.
> > > We will include this additional discussion as well as our results about $\epsilon$ in the updated manuscript. Thank you again for your comments and rebuttal response!

---

### Official Review · Reviewer_dD28 · 2025-03-18

**Overall Recommendation:** 3

**Summary:**

This paper focuses on evaluating neuron-level explanations in deep learning models, particularly in the context of mechanistic interpretability. While many existing methods generate textual explanations for individual neurons, a critical challenge remains: how to assess the quality and reliability of these explanations. To address this issue, the authors introduce NeuronEval, a unified mathematical framework that systematically organizes and compares 18 different evaluation metrics used in previous studies.

## update after rebuttal

**Claims And Evidence:**

"Input-based explanations" are vague. Some attribution-based methods also explain the input, but they are not related to neuron explanations.

**Essential References Not Discussed:**

N/A

**Experimental Designs Or Analyses:**

The experiments designed under the proposed framework are reasonable.

**Methods And Evaluation Criteria:**

The assessment methods used are reasonable.

**Other Comments Or Suggestions:**

Please see weaknesses, I am willing to raise my score if the author can address my concerns convincingly.

**Other Strengths And Weaknesses:**

**Strengths：**

1. The idea of using a unified theoretical framework to evaluate metrics is novel and meaningful, and this idea will have a profound impact on the design of future benchmarks.

2. The selected evaluation metrics are sufficient and can reflect the applicability of the framework.

**Weakness:**

1. In terms of experimental analysis, the paper lacks analysis of some evaluation metrics that have not passed the corresponding tests. Such analysis can help the reaction indicators evaluate the rationality of the experiment.
2. The paper discusses the value of a single metric, but existing work generally considers the performance of a combination of several metrics. Therefore, I think the paper needs to add some experiments to study how the combined metric fits into the proposed theoretical framework.
3. In the author's description in Section 3, we can see that the author has vaguely seen that the evaluation metrics can be classified, but the experimental results are only briefly mentioned without further analysis.
4. Neuro Explanation seems to be lacking if it does not consider the evaluation of the explanation of the parameters of the neurons in the intermediate layers, but only considers the explanation of the neurons in the last layer.
5. "Input-based explanations" are vague. Some attribution-based methods also explain the input, but they are not related to neuron explanations.

**Questions For Authors:**

NA

**Relation To Broader Scientific Literature:**

This framework has certain implications for the evaluation of existing black-box explanation work.

**Theoretical Claims:**

The theoretical part has some shortcomings.

---

> ### Author Rebuttal · Authors · 2025-04-01
>
> Thanks for the review!
>
> Please see https://drive.google.com/file/d/1OHMxyMW1KVIzxd2Rd_Hx34qIjVecUVmo/view for our new experimental results, in particular Tables G1-G4 as those experiments were conducted to address your questions. Below we address your concerns and questions in detail.
>
> > **Weakness 1 - Lacks analysis of some metrics that failed the tests**
>
> We would like to clarify that in Appendix C of the manuscript, we have conducted extensive theoretical analysis on why metrics fail the tests; and in our motivating example (Sec 4.1, table 2, Figure B.1) we showed why recall and precision fail the tests.
>
> In particular, failing the extra labels test is caused by the metric not being able to distinguish between the correct concept (pet) and a superclass (animal), while failing the missing labels test is caused by the metric not being able to differentiate the correct concept (pet) from a subclass (dog/cat). Overall, our tests are a mathematically formalized measurement of whether a certain metric can differentiate between a correct concept vs sub/superclass of it. For many metrics, this only happens on imbalanced neurons/concepts that do not activate on most inputs as we show in tables C1-C4, which indicates failure is tied to poor metric performance on imbalanced data. Please let us know if you had a specific kind of analysis in mind that you would like to see.
>
> > **Weakness 2 - Combinations of Metrics**
>
> Thank you for the suggestion! We would like to point out that our current result (Table 3, 4) already contains a combination of metrics, since F1-score is the harmonic mean of Recall and Precision: $F1 = 2/(recall^{-1} + precision^{-1})$. In the original manuscript, we also briefly discuss the idea of combining different metrics on lines 900-901 in the Appendix.
>
> Following your comment, we have conducted additional experiments evaluating more extensively whether combinations of metrics can work as a good evaluation metric. Specifically, inspired by F1-score, we used the harmonic mean of other existing metrics to see how they perform. Full results are shown in Tables G1-G4 in [our new results](https://drive.google.com/file/d/1OHMxyMW1KVIzxd2Rd_Hx34qIjVecUVmo/view). We can see that many combinations of metrics achieve quite a good performance, and now pass the theoretical missing/extra labels tests. For example the harmonic mean of Balanced Acc and Inverse Balanced Acc performed well. In general, our initial results indicate that combining a metric that passes the extra labels test with a metric that passes the missing labels test will pass both tests. Conversely, combining two metrics that fail the same test, such as Recall and AUC will still fail that test.
>
> > **Weakness 3 - Section 3: Evaluation Metrics can be classified**
>
> We are not sure which specific part the reviewer is referring to. Do you refer to framing neuron explanation as a binary classification problem in line 114? If so, we have discussed the details in Appendix A.2. We would be happy to discuss further if you have specific questions.
>
> > **Weakness 4 - Intermediate layer explanations**
>
> We believe there might be some misunderstanding, our results and contributions are not limited to final layer neurons. Our proposed NeuronEval framework in Sec 3 and the theoretical missing/extra labels test (Tab. 3, Fig. 2, App. C) cover explanations of *all* neurons
> regardless of where they are, including hidden layers neurons, final layer neurons and even non-neuron units like directions in activation space.
> In addition, we did show intermediate or non final-layer results in our experiments, including:
>  - intermediate layer neurons (settings 2 & 4, Tab F1-F2),
>  - concept neurons in an intermediate layer of a concept bottleneck model (setting 5 in Tab. F3 & setting 7 in Tab. F.6)
>  - linear probes trained on hidden layer representations (setting 6 in Tab. F3 and setting 8 in Tab. F6).
>
> Our main text results (Table 3, Table 4) are averaged across settings and contain these results.
>
> > **Weakness 5 - “Input-based explanations” are vague**
>
> Thank you for pointing this out. We will change the term to *Input-based neuron explanations* to avoid confusion.
>
> > **Theoretical claims - the theoretical part has some shortcomings.**
>
> Could the reviewer expand on what specific shortcomings in the theoretical part are? We would be happy to discuss and clarify further if you have any specific comments on shortcomings.
>
> > **Summary**
>
> We believe that we have addressed your major concerns by clarifying our theoretical analysis and experiment results in the intermediate layer neurons (Weakness 1, 4), running additional experiments on combined metrics (Weakness 2) and clarifying our terminology (Weakness 5). We had a hard time understanding a few concerns (Weakness 3, Theoretical Claims), and we hope the reviewer can clarify these if concerns still remain. Otherwise, we hope you consider adjusting the rating if our response has addressed your concerns.

---

> > ### Comment · Reviewer_dD28 · 2025-04-07
> >
> > Thanks for the author's detailed reply. I think most of the concerns have been clarified, so I decided to increase my score.

---

> > > ### Author Response · Authors · 2025-04-07
> > >
> > > Thanks again for the review and response! We are happy to hear we have addressed your concerns.

---

### Decision · Program_Chairs · 2025-05-01

**Decision:**

Accept (poster)

**Comment:**

This paper studies how to evaluate neuron explanations. The authors propose NeuronEval, a framework for the meta-evaluation of input-based explanation metrics. Given a textual explanation of an input, various metrics have been proposed to assess how faithfully the explanation captures neuron behavior. NeuronEval unifies these existing evaluation methods and characterizes their differences by varying factors such as the evaluation metric, the source of the concept vector, the granularity of activation vectors, and the probing dataset.

Overall, reviewers found the unified framework meaningful and considered this work a valuable and timely contribution to the field of interpretability.